# Morphometric evaluation of the anterior cranial fossa during the prenatal stage in humans and its clinical implications

**Wojciech Derkowski** [1] *, **Alicja Kędzia**[2], **Krzysztof Dudek**[3], **Michał Glonek**[4]

1 University of Opole, Faculty of Health Sciences, Opole, Poland, 2 Wroclaw Medical University, Wroclaw, Poland, 3 Faculty of Mechanical Engineering, Wrocław University of Science and Technology, Wrocław, Poland, 4 Specialized Neurological Practice, Neurological Office, Chrząstowice, Poland

* w.derkowski@hipokrates.org

## Abstract

The study examines the morphometric development of the anterior cranial fossa in human fetuses and its clinical implications. The anterior cranial fossa, crucial for protecting the frontal lobes, was analyzed during prenatal development using innovative computer image processing techniques. We hypothesized that the growth of the anterior cranial fossa is not uniform throughout fetal development and that changing geometric relationships are important for possible therapeutic interventions in cases of congenital defects. A metrological assessment was conducted on 77 fetuses, aged 10 to 27 weeks of gestation, to investigate developmental patterns, including symmetry, sexual dimorphism, and structural changes relative to other cranial fossae. Key findings revealed a decrease in the anterior cranial fossa angle, compensated by an increase in the middle cranial fossa angle. Symmetry in cranial base development was observed, and sexual dimorphism was evident, with male fetuses showing larger angles and females displaying greater height of the crista galli. These results were discussed in the context of existing anatomical and imaging studies. Clinically, the findings provide insights into the pathomechanism of congenital skull and brain defects, supporting the potential for early diagnostic and therapeutic interventions. Our study leads to the conclusion that the growth of the anterior cranial fossa is not uniform; in the first trimester, allometric growth occurs, while at the same time the angle of the anterior cranial fossa decreases and its depth increases towards the middle cranial fossa. In the second trimester, growth continues but becomes more uniform, with only minor changes in the angle of the anterior cranial fossa. There is a gradual decrease in the angle between the smaller wings of the sphenoid bone as the depth of the anterior cranial fossa increases in the frontal plane. Sexual dimorphism is visible in the area of the anterior cranial fossa already in the prenatal period.

## 1. Introduction

Congenital defects of the central nervous system are among the most severe, often affecting lifelong development. Advances in prenatal diagnostics, particularly ultrasound, have

**Data Availability Statement:** The minimal data are within the article and its Supporting Information files.

**Funding:** The author(s) received no specific funding for this work.

**Competing interests:** The authors have declared that no competing interests exist.

improved early detection, increasing the potential for in utero interventions. Precise analysis of skull and intracranial development is vital, as it may provide insights into conditions such as hydrocephalus and cranial suture disorders. Consequently, a precise analysis of the individual development of the skull and intracranial structures in humans has become increasingly important. Advancements in metrological techniques have enabled more accurate measurements of the developing fetus. The use of modern computer methods allows for obtaining more information from analyzed brain cross-sectional images.

The anterior cranial fossa, which supports the frontal lobes, plays a critical role in brain development. Abnormalities in this region can result in motor and cognitive impairments, making its study essential. During prenatal development, the cranial base angle—formed between the nasion and basion points—changes significantly, reflecting the shift toward an upright posture. This angle is useful in assessing both normal and pathological development.

During normal fetal development, the cranial base angle initially approaches 180°, then gradually changes in value, reflecting the shift in the orientation of the foramen magnum axis to the body's axis, corresponding to the upright posture characteristic of humans. Studies of the cranial base angle in newborns with cleft palate show no significant difference compared to healthy children, in contrast to the length of the clivus, which is shorter in children with congenital cleft palate [1]. The initial stages of skull development are related to brain development, with the first notochordal mesenchymal condensations appearing on the 28th day post-fertilization. Subsequently, skull cartilages form, initially mainly in the posterior fossa region. By the 7th week of fetal life, the cartilaginous cranial base is well developed. At that time, the cranial base angle is set up and changes little during further head growth [2]. During fetal development, all external and internal dimensions of the skull increase several times, but not uniformly. Between the 10th and 40th weeks of development, the linear dimensions of the skull, the brain, and the anterior fossa structures increase 6 to 7 times, while the linear dimensions of the posterior fossa structures increase only 4 to 5 times [3]. Skull formation progresses through phases: membranous (desmocranium), cartilaginous (chondrocranium), and bony (osteocranium) [4].

At the turn of the 2nd and 3rd weeks of fetal life, induction occurs at the anterior end of the embryonic disc. Desmocranium becomes the precursor of the dura mater. In areas of greater curvature of the brain surface, the membranous skull (desmocranium) hardens. Thus, bands of dura mater condensation form between the frontal and temporal lobes, between the occipital lobe and cerebellum, and between the two cerebral hemispheres [5].

At the base, cartilage formation begins near the bands of dura mater condensation. The first primary cartilages form in the posterior cranial base and then spreads forward. The peak development period of the cartilaginous skull occurs in the 3rd month of development [6].

Detailed morphological studies of the cartilaginous skull of an 8-week-old fetus show that it forms a continuous mass of well-developed cartilage, transitioning into forming cartilages in the nasal, orbital, and ear regions around the future round foramen and the parietal lamina area [6]. The anterior cranial fossa's growth is linked to brain development, and ossification occurs in a well-defined sequence. The development of the anterior cranial fossa is particularly interesting concerning the pathogenesis of premature cranial suture closure. The anterior fossa expands in width at the presphenoid synchondrosis, in the ethmoid cartilage, and the frontal suture—in the midline. Longitudinal growth occurs at the intersphenoid synchondrosis, ethmoid cartilage, sphenoethmoidal and sphenoid-frontal synchondrosis, and the sphenoid-frontal suture. This growth results in the displacement of the ethmoid and frontal bones relative to the sphenoid bone [6].

Detailed morphological studies of the chondral skull of an 8-week-old fetus show that it forms a continuous mass composed of well-developed cartilage tissue, transitioning into the

newly forming cartilages of the nasal, orbital, and ear regions around the future round foramen and around the parietal plate [7]. The anterior base of the chondral skull subsequently undergoes ossification, with the appearance of appropriate ossification centers. The growth of skull bones can occur in two ways: the need for bone displacement forces growth at the site of sutures or synchondroses, while the pressure of the cerebral cortex activates the periosteum on the inner and outer surfaces of the skull. The outer periosteum causes the deposition of bone tissue and an increase in bone thickness from the outside—an action of osteoblasts, while the inner periosteum induces bone resorption from the inside—an action of osteoclasts.

The development of the anterior base of the skull is particularly interesting in terms of the pathogenesis of premature cranial suture closure. The anterior base expands in width at the site of the presphenoid synchondrosis, the ethmoid cartilage, and the frontal suture—in the midline. In contrast, length growth occurs at the intersphenoid synchondrosis, the ethmoid cartilage, the sphenoethmoidal and sphenoidal synchondrosis, and the sphenoidal-frontal suture. As a result of this growth, the ethmoid and frontal bones are displaced relative to the sphenoid bone [6].

The issue of symmetry during the development of the anterior base of the human skull is interesting. The question of symmetry and lateralization of functions in both brain hemispheres has been the focus of many researchers in recent years. Planimetric measurements of temporal planes during the prenatal period showed a size advantage on the left side. In right-handed adults, the left-brain hemisphere is dominant and responsible for verbal, ideational, analytical, arithmetic, and computational functions. In contrast, the right hemisphere predominates in functions such as image and pattern creation, geometric and spatial imagination, musical, and holistic processing. The problem of symmetry of the lateral brain ventricles in the fetal period has been considered based on ultrasound studies during pregnancy [7]. Ultrasound results in utero led to the conclusion that a certain degree of asymmetry of the lateral ventricles occurs in some fetuses. Probably, isolated asymmetry of the lateral ventricles has no clinical significance and should not be considered pathological.

As the skull develops, symmetry and sexual dimorphism emerge, with implications for congenital defects like holoprosencephaly, corpus callosum disorders, and craniosynostosis [8].

The group of neuronal migration disorders includes cortical defects caused by impaired migration of neuroblasts from the periventricular matrix to the cerebral cortex. These include: agyria or lissencephaly (smooth brain), pachygyria (broad gyri), polymicrogyria, heterotopias of the gray matter, and porencephalia vera (true porencephaly) or schizencephaly. These lead to mental and motor retardation to varying degrees and can cause epileptic seizures. Among infratentorial structure defects, we include Chiari syndrome [9–11] and Dandy-Walker syndrome. Other developmental defects include hydrocephalus, microcephaly, macrocephaly or anencephaly [12]. Hydrocephalus often coexists with other congenital defects, such as spina bifida, encephalocele, Dandy-Walker syndrome, holoprosencephaly, or agenesis of the corpus callosum. The mortality rate among fetuses with hydrocephalus depends on the accompanying non-central nervous system defects. The degree of neurological deficit after birth depends on the severity of hydrocephalus and the accompanying intracranial defects [13].

Finally, we cannot forget about the cranial bone defects such as Crouzon syndrome (craniofacial dysostosis). These are most often caused by premature closure of cranial sutures (craniosynostosis) [14], which can lead to cranial constriction (craniostenosis). Several types of cranial constriction can be distinguished: oxycephaly, scaphocephaly, acrobrachycephaly, and plagiocephaly. Other bone defects include Apert syndrome, Pierre-Marie-Sainton syndrome (cleidocranial dysostosis) [15] and Grieg's primary hypertelorism. Considering the etiology of congenital nervous system defects, both genetic and environmental factors, such as chemical

substances taken by the pregnant woman, especially alcohol [16], ionizing radiation, viral infections during pregnancy, and nutritional factors [17, 18], must be considered.

This study aims to assess the anterior cranial fossa during prenatal development using modern image processing techniques. Details aims included:

1. Measuring the anterior cranial fossa at different developmental stages and describing the relationship between specific dimensions and developmental age.

2. Tracking the developmental links of the anterior cranial fossa in comparison to the other two cranial fossa.

3. Analyzing the symmetry of the skull base during development.

4. Investigating potential differences in the anterior cranial fossa and skull base between male and female fetuses during the prenatal period.

5. Clinical considerations: using the results to create a mechanical model of the developing skull for analyzing the causes of congenital skull and brain defects and exploring potential treatment options.

The study's hypothese was that the growth of the anterior cranial fossa is not uniform throughout fetal development and that changing geometric relationships are important for possible therapeutic interventions in cases of congenital defects.

## 2. Material and methods

### 2.1. Material characteristics

The study material consisted of 77 human fetuses aged between 10 and 27 weeks of gestational age (mean = 20, SD = 3.2) and crown-rump length ranging from 55 to 260 mm (mean = 184, SD = 41). The fetuses, fixed using conventional formalin solutions, were subsequently dissected. A horizontal subbasal incision separated the cranial vault, the brain was removed, and then the specimen of the anterior cranial fossa was positioned in a custom-designed stand and appropriately illuminated. The image was then captured using a Sony video camera and a special image processing card from the ELF system. Images were saved on the computer's hard drive and further processed using ELF and Scion Image for Windows image analysis software. The morphological age of the fetuses was estimated based on the crown-rump length (v-tub) using an empirical relationship derived from the Scammon and Calkins nomogram, where: *v-tub* [cm]–crown-rump length (CR), age [months]–fetal age. The fetal ages ranged from 10.5 to 25.9 weeks of gestational age. The difference in fetal ages between males and females in the study material was not statistically significant. The fetuses came from the collections of the local anatomical museum. The data (human fetuses) were accessed for research purposes from October 27, 2000, after obtaining ethics approval for this study of Bioethics Committee at the Akademia Medyczna we Wrocławiu No. KB-540/2000. Authors had not access to information that could personally identify individual participants during or after data collection. The need for consent was waived by the ethics committee. Bioethics Committee at the Akademia Medyczna we Wrocławiu (currently the Medical University of Wrocław) after familiarizing themselves with the research project and the documents submitted with the application decided to consent to conducting the study at the Department of Normal Anatomy of the Akademia Medyczna we Wrocławiu.

### 2.2. Preparation method and research setup

The research setup consisted of a dissection table and a custom-designed stand for precise horizontal sectioning of the specimen. Positioned above the stand was a Sony video camera (also mounted on a stand with centimeter scale), used for image acquisition. The image was then transferred to a computer via an LFG468/UVC image processing card and subsequently

subjected to processing and metrological analysis using ELF and Scion Image for Windows software. In comparison to previous anatomical studies of fetal brains reported in the literature, the novelty of this study lay in the method of digital image acquisition, processing, and analysis, which allowed for the creation and metrological analysis of a significantly larger number of images in a much shorter time than conventional photography methods, for example.

### 2.3. Anthropometric measurements

Anthropometric measurements were conducted using a ruler, an electronic caliper with a digital display, and a flexible centimeter measuring tape. The following measurements were taken:

- Crown-rump length

- Transverse head dimension

- Longitudinal head dimension

- Head circumference

- Chest circumference.

### 2.4. Computer image analysis

One of the programs used was Scion Image for Windows (Fig 1). The study also used ELF v. 4.2., which works in conjunction with a special image acquisition card (Fig 2).

### 2.5. Statistical analysis

The study results were analyzed and then presented in descriptive, tabular and graphic form. The statistical software Statistica v.13.3 (TIBCO Software Inc., Palo Alto, CA, USA) was used in calculations. For categorical and continuous variables, means (M), standard deviations (SD), medians (Me), lower quartiles (Q1) and upper quartiles (Q3) and values such as the smallest (Min) and the largest (Max) values were calculated. Nominal and ordinal variables are presented in tables as numbers (n) and proportions (%). The compliance of the empirical distributions of quantitative variables with the theoretical normal distribution was verified using the Shapiro–Wilk test. Bartlett's test was used to check the homogeneity of variance of the results. The significance of differences in average values (medians) of continuous variables with non-normal distribution or with heterogeneous variances in two independent groups was verified with the Mann–Whitney U test. The Wilcoxon test was used to assess the

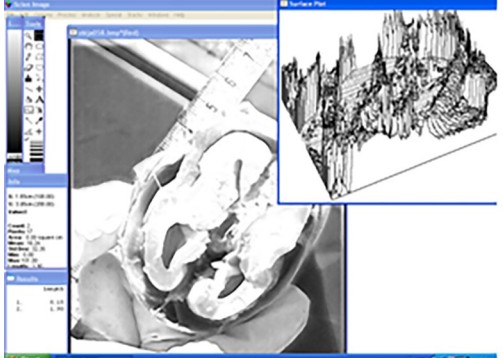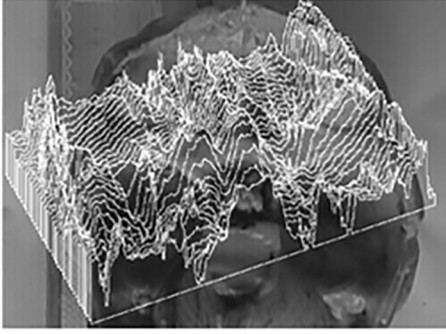

**Fig 1.** Main window and auxiliary windows of Scion Image for Windows (left); surface plot of the skull base (right).

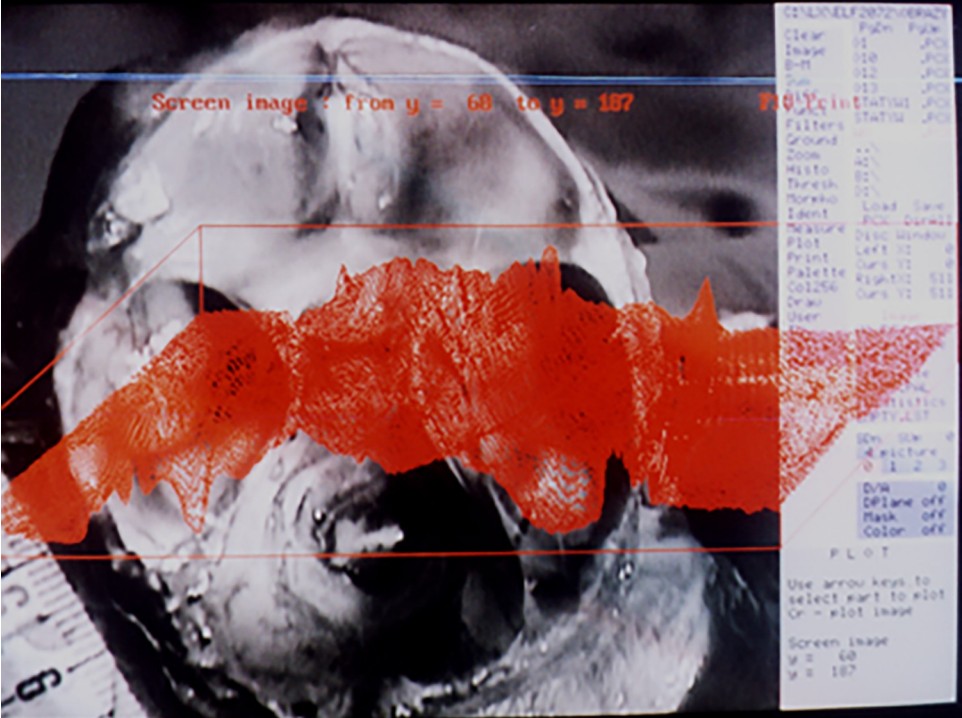

**Fig 2. The red color on the image shows the surface plotter in the highlighted region of the skull base.** ELF.

symmetry of dimensions on the left and right side of the head. For each feature, a correlation diagram was then plotted showing the relationship between the measurement feature and fetal age [19]. All statistical hypotheses were verified by two-sided tests at the significance level of p < 0.05. Conclusions from the analysis were supplemented with the interpretation of the effect size index value–a statistical measure used to assess the scale (size) of the obtained effect, e.g. the difference between groups or the strength of relationships between variables. For the coefficient of determination, the division into weak effect ($R^2 < 0.02$), moderate effect ($R^2 < 0.13$) and strong effect ($R^2 > 0.26$) was adopted according to Ellis [20].

The fetal autopsy specimens had been collected for several years. Not all of them were complete. Therefore, the database also contained missing values. The total number of missing values did not exceed 21%. Missing data were replaced by stochastic regression imputation. Multiple imputation creates several versions of complete data sets by replacing missing values with probable (reliable) values. m = 5 data sets were assumed. The data sets were identical in terms of observed (measured) values but differed in imputed (missing) values. Parameters of interest (means, SDs, and correlation coefficients) were then estimated from each imputed data set. Finally, for each anthropometric parameter, the average value of m estimates was calculated, and its variance was determined. Imputations was performed by R package mice (Multivariate Imputation by Chained Equations) [21]. Due to multiple comparisons, *p*-values in the tables were adjusted using the Holm-Bonferroni correction.

## 3. Results

### 3.1.a. General characteristics of measurement features

The fetal development characteristics studied can be divided into:

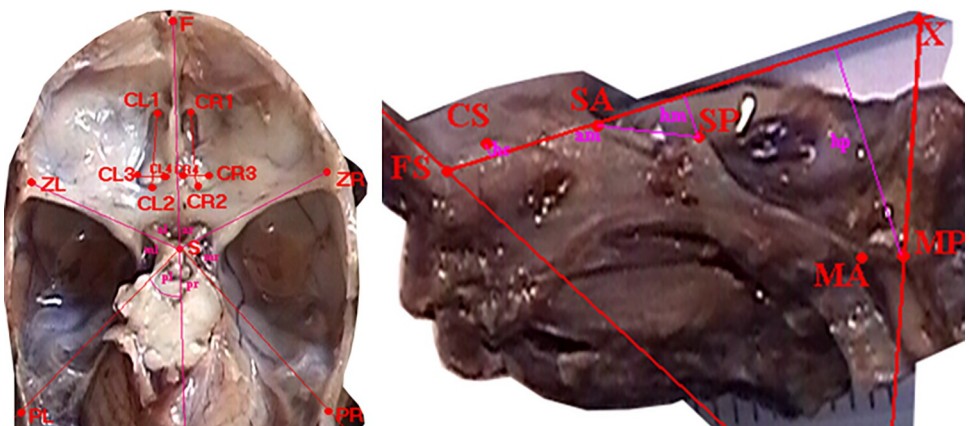

**Fig 3.** Skull base from above (left); sagittal section in the midline plane (right), measurement points marked.

- Measurement features characterizing the fetus, obtained from anthropometric measurements.

- Distances between specified measurement points in the anterior cranial fossa and other cranial fossae, obtained using computer image analysis programs.

- Derived features from measured distances, particularly values of specified angles characterizing skull geometry.

### 3.1.b. Characteristics of measurement points

To perform distance measurements on images of anatomical preparations uploaded to the computer, fixed measurement points were established (Figs 3, 4):

- F–*nasion* point

- S–center of the *sella turcica*

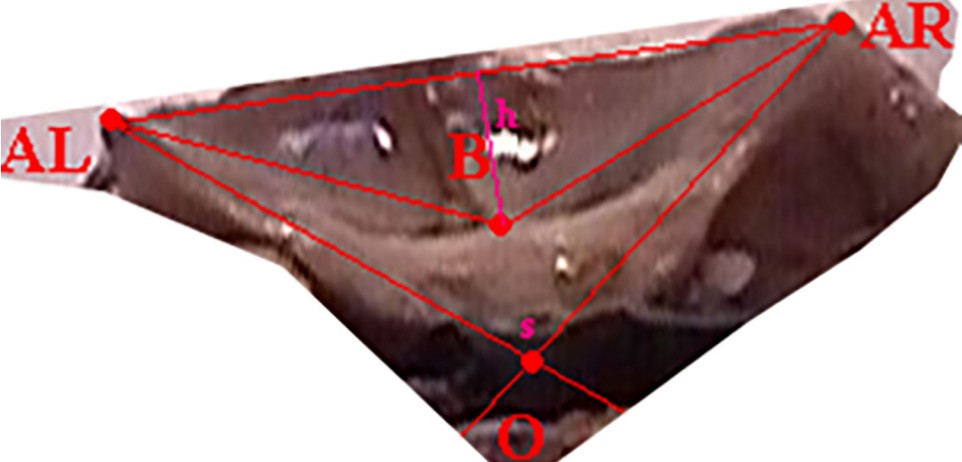

**Fig 4. View of the anterior fossa of the skull from behind—projection on the frontal plane, measurement points.**

- ZL–left zygomatic bone ossification point

- ZR–right zygomatic bone ossification point

- PL–point on the temporal bone corresponding to the center of attachment of the left ear cartilage

- PR–point on the temporal bone corresponding to the center of attachment of the right ear cartilage

- CL1 –center of the anterior edge of the left part of the cribriform plate

- CL2 –center of the posterior edge of the left part of the cribriform plate

- CL3, CL4 –points defining the transverse dimension of the left part of the cribriform plate

- CR1 –center of the anterior edge of the right part of the cribriform plate

- CR2 –center of the posterior edge of the right part of the cribriform plate

- CL3, CL4 –points defining the transverse dimension of the right part of the cribriform plate

- MA–*basion* point (on the anterior edge of the foramen magnum)

- MP–*opisthion* point (on the posterior edge of the foramen magnum)

- FS–point in the midline plane at the intersection of the floor of the anterior cranial fossa and a line perpendicular to the frontal bone

- CS–highest point of the sagittal crest

- SP–highest point of the slope

- SA–point on the anterior edge of the squamous part of the temporal bone

- X and Y–auxiliary points, plotted so that point X lies on the line FS-SA, line FS-Y is perpendicular to the frontal bone, and line X-Y is perpendicular to the occipital bone at point MP

- AL–point on the left lesser wing of the sphenoid bone constituting its lateral vertex

- AR–point on the right lesser wing of the sphenoid bone constituting its lateral vertex

- O–intersection point of tangents to the upper surfaces of the lesser wings of the sphenoid bone at points AL and AR

- B–lowest point of the floor of the anterior cranial fossa

### 3.1.c. Characteristics of specified angles and distances, characterizing skull geometry

- longitudinal dimension of the anterior cranial fossa (S-F)

- the angle of anterior cranial fossa (ZL-S-ZR = al + ar)

- the depth of the anterior cranial fossa (h)

- the angle between the lesser wings of the sphenoid bone (s)

- the depth of the posterior cranial fossa (hp)

- the angle of the skull base (FS-SA-SP)

- the depth of the middle cranial fossa (hm)

- the height of the crest of the sphenoid bone (hc).

## 3.2. Measurement values

Anthropometric measurements allowed us to determine the following values characterizing the examined fetuses: vertex-tuberale length (*v-tub*), transverse dimension of the head (*eu-eu*), longitudinal dimension of the head (*g-op*), head circumference (*HC*) and chest circumference (*CC*). Basic statistics of anthropometric parameters in the group of male and female fetuses and the results of significance tests (Mann-Whitney U test) are presented in Table 1.

The test results show that the difference in anthropometric dimensions between female and male fetuses in the analyzed group is not statistically significant.

We also investigated how anthropometric dimensions change with fetal age. For this purpose, correlation diagrams were prepared (Table 2). To estimate the growth rate of anthropometric parameters, segmented regression was used. The growth model of fetal dimensions consisting of two simple regressions proved to be sufficiently accurate. The value of the coefficient of determination for the estimated models exceeded 0.67, i.e. models based on morphological age explain at least 67% of the variability of somatic characteristics of fetuses.

**Table 1.** Descriptive statistics of age and anthropometric parameters of 77 human fetuses.

| Parameters | All<br>N = 77 | Male<br>N = 47 | Female<br>N = 30 | Male *vs* Female |
|---|---|---|---|---|
| *Age* (weeks) | | | | Z = 1.410<br>p = 0.158 |
| *M (SD)* | 20.0 (3.2) | 20.2 (3.6) | 19.7 (2.5) | |
| *Me [Q1; Q3]* | 20 [18; 22] | 21 [18; 22] | 20 [18; 21] | |
| *Min—Max* | 10.5–25.9 | 10.5–25.9 | 13.0–24.5 | |
| *v-tub* (mm) | | | | Z = 1.441<br>p = 0.149 |
| *M (SD)* | 184 (41) | 187 (46) | 180 (31) | |
| *Me [Q1; Q3]* | 187 [161; 205] | 192 [162; 210] | 181 [159; 200] | |
| *Min—Max* | 55–260 | 55–260 | 95–235 | |
| *eu-eu* (mm) | | | | Z = 1.007<br>p = 0.268 |
| *M (SD)* | 49 (9) | 49 (10) | 48 (6) | |
| *Me [Q1; Q3]* | 50 [45; 54] | 51 [45; 55] | 48 [45; 51] | |
| *Min—Max* | 18–69 | 18–69 | 34–60 | |
| *g-op* (mm) | | | | Z = 0.272<br>p = 0.786 |
| *M (SD)* | 63 (11) | 63 (13) | 64 (9) | |
| *Me [Q1; Q3]* | 65 [59; 70] | 67 [59; 70] | 63 [58; 71] | |
| *Min—Max* | 22–86 | 22–86 | 44–85 | |
| *HC* (cm) | | | | Z = 0.000<br>p = 1.000 |
| *M (SD)* | 179 (31) | 177 (36) | 182 (22) | |
| *Me [Q1; Q3]* | 181 [170; 200] | 185 [170; 198] | 179 [165; 203] | |
| *Min—Max* | 59–245 | 59,0–245,0 | 131–225 | |
| *CC* (cm) | | | | Z = 1.541<br>p = 0.123 |
| *M (SD)* | 163 (29) | 165 (33) | 161 (21) | |
| *Me [Q1; Q3]* | 171 [151; 180] | 175 [152; 185] | 163 [145; 177] | |
| *Min—Max* | 57–210 | 57–210 | 105–197 | |

*v-tub*—crown-rump length of the body, *eu-eu*—transverse dimension of the head

*g-op*—longitudinal dimension of the head, *HC*—head circumference, *CC*—chest circumference, Z - significance test statistic (Mann-Whitney), *p*—test significance level

**Table 2. Coefficients of the segmented regression model of linear parameter changes as a function of fetal age: First segment:** $Y = a_0 + a_1 * Age$ (*Age* from 10 weeks to $Age_{cut-off}$), second segment: $Y = b_0 + b_1 * Age$ (*Age* from $Age_{cut-off}$ to 26 weeks), $a_0$ –first segment constant, $a_1$ - slope of the first segment, $b_0$ –second segment constant, $b_1$—slope of the second segment), $R^2$—coefficient of determination.

| Parameter $Y$ | $Age_{cut-off}$ (week) | $a_0$ (cm) | $a_1$ (cm/week) | $b_0$ (cm) | $b_1$ (cm/week) | $R^2$ |
|---|---|---|---|---|---|---|
| *v-tub* (mm) | 18.0 | -80.9 | 13.43 | -65.4 | 12.50 | 0.998 |
| *eu-eu* (mm) | 19.3 | -13.7 | 3.31 | 39.2 | 0.59 | 0.666 |
| *g-op* (mm) | 18.0 | -22.0 | 4.62 | 34.0 | 1.55 | 0.704 |
| *HC* (mm) | 19.4 | -52.8 | 12.45 | 189.6 | 0.08 | 0.689 |
| *CC* (mm) | 17.8 | -54.0 | 11.53 | 36.0 | 6.49 | 0.880 |

*v-tub*—vertex-heel length, *eu-eu*—transverse dimension of the head, *g-op*—longitudinal dimension of the head, *HC*—Head circumference, *CC*—Chest circumference

In the studied material, the growth rate of all anthropometric parameters (except *v-tub*) was higher until approximately week 18 and then decreased (Fig 5). Basic descriptive statistics of the measured distances between the identified measurement points in the horizontal plane are presented in S1 Appendix.

A statistically significant difference between male and female fetuses was observed in the value of the anterior cranial fossa angle (*a*) and height of the crest of the sphenoid bone (*hc*) (Table 3). In male fetuses, the anterior cranial fossa angle was on average 10° larger (146° *vs.* 136°, $p = 0.002$) and the sphenoid crest height was 0.5 mm smaller (1.5 mm *vs.* 2.0 mm, $p = 0.007$, Fig 6).

Analyzing scatterplots of cranial fossa parameters in relation to fetal age, their uneven course was observed. Up to about 18 weeks of age, the rate of growth of linear parameters is higher, and then it decreases (Fig 7). In the case of angular dimensions, the opposite is true. The parameters of segmental models are presented in Table 4. Regression coefficients *R* different from zero at the level of $p < 0.05$ are **_marked._**

Up to 18 weeks of fetal life, the longitudinal dimension of the anterior cranial fossa (S-F) increases at a rate of 1.93 mm/week, and from 18 weeks onwards it decreases and increases at a rate of 1.09 mm/week. The depth of the anterior cranial fossa changes similarly. Its growth rate is 0.05 mm/week up to 17.7 weeks, and then decreases to 0.02 mm/week. The nature of the increase in the depth of the posterior cranial fossa and the depth of the middle cranial fossa is similar, but the differences in the inclination of both segments are not statistically significant. The changes in the angular dimensions have the opposite nature. Up to week 18, the alpha angle decreases at a rate of 3.5° per week, then slowly increases at a rate of 1.3° per week. Similarly, the *s* angle decreases at a rate of 6.2°/week until week 16.6, and then slowly increases at a rate of 0.5°/week (Fig 8). The *hc* depth and *FS-SA-SP* angle did not change significantly from week 10 to week 27 ($p > 0.05$).

Sexual dimorphism of dimensions and auxiliary angles characterizing the cranial fossae was analyzed by comparing the mean values in the group of male and female fetuses using non-parametric tests. Statistically significant differences were observed for the angle of the anterior cranial fossa on both the left and right sides. These angles were larger in male fetuses (71° *vs.* 67°, p = 0.007 and 74° *vs.* 69°, $p = 0.007$, respectively).

The results of the symmetry analysis of the parameters characterizing the cranial fossae are presented in Table 5. On the right side, the distances from the center of the pituitary gland to the zygomatic bone and from the nasion point to the zygomatic bone as well as the angle of the anterior cranial fossa were greater.

In the analyzed group of fetuses, 7 fetuses were diagnosed with brain malacia- "clinically silent" brain infarction (SBI). There was a statistically significant relationship between SBI and

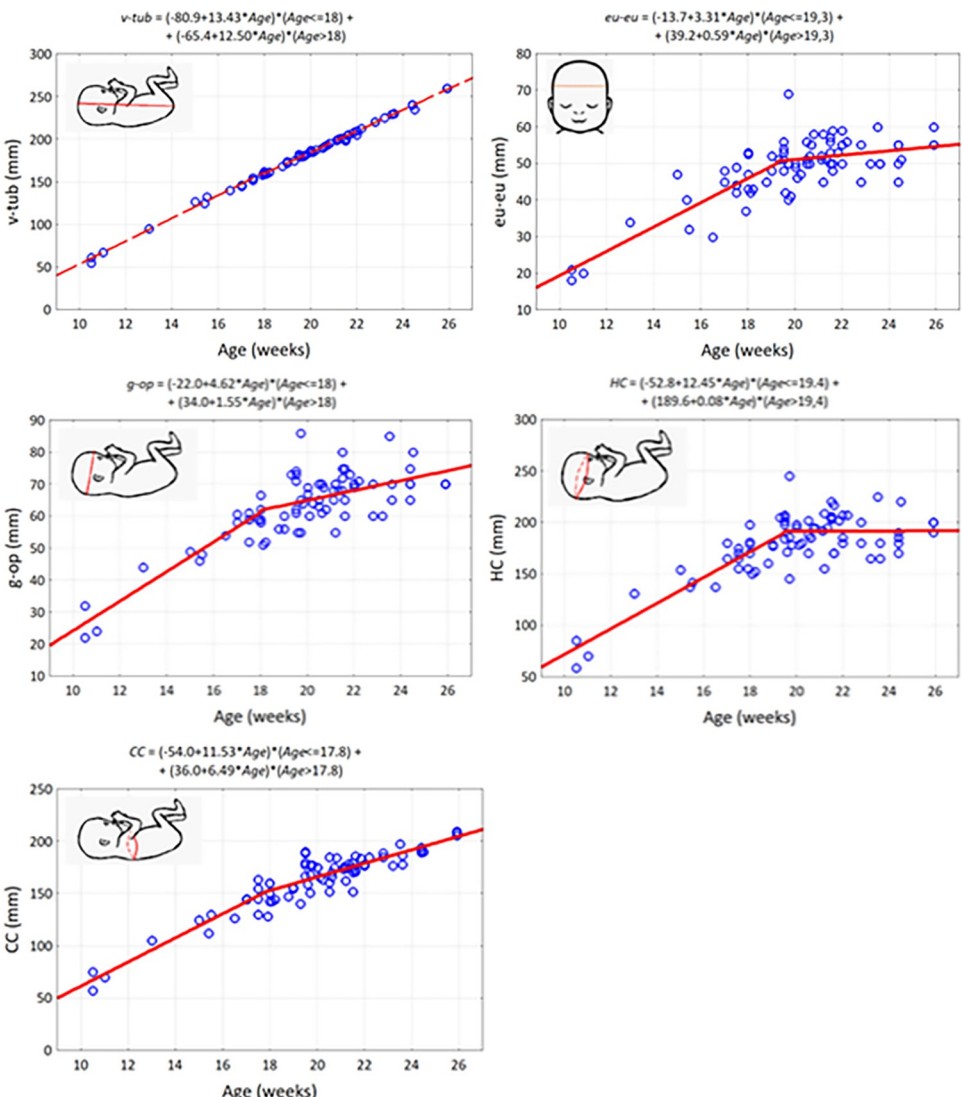

**Fig 5. Scatterplots of anthropometric parameters of 77 human fetuses as a function of age and parameters of two-segment regression lines.**

dimensions and dimension proportions. It turned out that the a/p angle ratio may be such a predictor. Below is a figure illustrating how the a/p angle ratio changes with age (it can be assumed that for fetuses without SBI it does not change significantly). The relative risk of SBI in fetuses with an a/p ratio > 1.52 is high and amounts to RR = 4.17 [1.14–15.3].

We performed an analysis of covariance (ANCOVA), where the dependent variable was the a/p angle ratio and the grouping factor was SBI, while the covariates were fetal age and gender. It turned out that in the group of fetuses with brain malacia, the age-adjusted a/p ratio is significantly higher (at $p < 0.00001$), which confirms that the a/p ratio can be used to assess the risk of brain malacia (Fig 9).

### 3.3. Growth of the anterior cranial fossa

The measurements conducted show an overall increase in all dimensions of the skull with age. However, the rate of growth is not uniform across all regions of the skull. The asynchronous

**Table 3. Descriptive statistics of the cranial fossa parameters of 77 fetuses and comparison results.**

| Parameters | All N = 77 | Male N = 47 | Female N = 30 | M vs F |
|---|---|---|---|---|
| Longitudinal dimension of the anterior cranial fossa, S-F (mm) | | | | Z = -0.005 p = 0.996 |
| M (SD) | 26.7 (5.3) | 26.4 (5.7) | 27.2 (4.6) | |
| Me [Q1; Q3] | 28 [24; 30] | 28 [24; 31] | 28 [23; 30] | |
| Min – Max | 9 – 36 | 9 – 34 | 17 – 36 | |
| Angle of the anterior cranial fossa, a (°) | | | | Z = 3.081 **p = 0.002** |
| M (SD) | 140.8 (13.0) | 144.4 (13.0) | 135.2 (11.1) | |
| Me [Q1; Q3] | 141 [133; 149] | 146 [134; 153] | 136 [126; 141] | |
| Min – Max | 113 – 181 | 113 – 181 | 113 – 156 | |
| Depth of the anterior cranial fossa, h (mm) | | | | Z = -0.168 p = 0.867 |
| M (SD) | 0.73 (0.15) | 0.72 (0.15) | 0.74 (0.15) | |
| Me [Q1; Q3] | 0.8 [0.6; 0.8] | 0.8 [0.6; 0.8] | 0.8 [0.6; 0.8] | |
| Min – Max | 0.2 – 1.1 | 0.2 – 1.0 | 0.5 – 1.1 | |
| Depth of the posterior cranial fossa, hp (mm) | | | | Z = -0.204 p = 0.839 |
| M (SD) | 1.91 (0.32) | 1.89 (0.33) | 1.93 (0.33) | |
| Me [Q1; Q3] | 1.9 [1.7; 2.1] | 1.9 [1.7; 2.1] | 1.9 [1.7; 2.2] | |
| Min – Max | 1.2 – 2.9 | 1.2 – 2.9 | 1.5 – 2.8 | |
| Depth of the middle cranial fossa, hm (mm) | | | | Z = -0.470 p = 0.839 |
| M (SD) | 4.43 (1.55) | 4.33 (1.49) | 4.59 (1.66) | |
| Me [Q1; Q3] | 4.4 [4.0; 5.1] | 4.5 [3.0; 5.0] | 4.3 [4.0; 6.0] | |
| Min – Max | 0.0 – 8.0 | 0.0 – 8.0 | 0.0 – 8.0 | |
| Height of the crest of the sphenoid bone, hc (mm) | | | | Z = -2.679 **p = 0.007** |
| M (SD) | 1.75 (0.88) | 1.53 (0.83) | 2.10 (0.87) | |
| Me [Q1; Q3] | 2.0 [1.0; 2.0] | 1.5 [1.0; 2.0] | 2.0 [1.5; 2.0] | |
| Min – Max | 0.0–4.0 | 0.0–4.0 | 1.0–4.0 | |
| Angle between the lesser wings of the sphenoid bone, s (°) | | | | Z = 0.877 p = 0.380 |
| M (SD) | 104.3 (1.5) | 105.5 (11.2) | 102.3 (12.0) | |
| Me [Q1; Q3] | 106 [96; 108] | 106 [99; 108] | 103 [93; 110] | |
| Min – Max | 79 – 108 | 84 – 142 | 79 – 130 | |
| Skull base angle, FS-SA-SP (°) | | | | Z = 0.460 p = 0.646 |
| M (SD) | 151.6 (10.6) | 152.2 (5.9) | 150.8 (12.3) | |
| Me [Q1; Q3] | 152 [149; 158] | 152 [149; 158] | 151 [148; 158] | |
| Min – Max | 117 – 180 | 122 – 180 | 117 – 180 | |

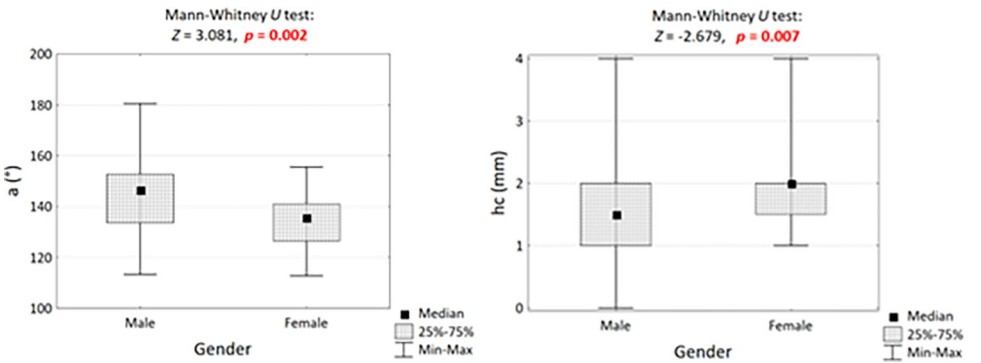

**Fig 6. Anterior cranial fossa angle in sex-differentiated fetal groups and significance test result.**

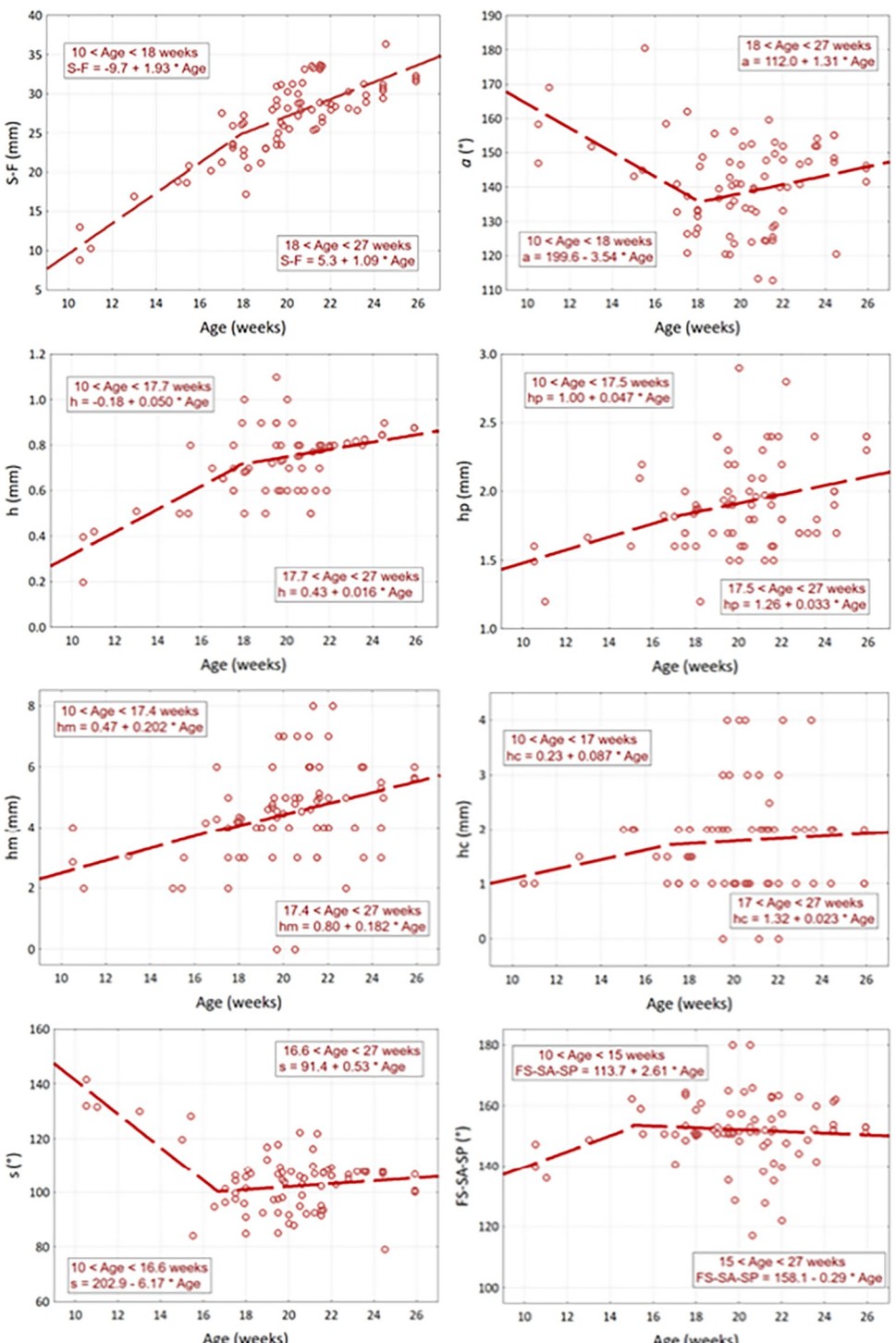

**Fig 7. Scatter plot of cranial fossa parameters depending on the age of 77 fetuses and the parameters of two regression line segments.**

enlargement of the anterior cranial fossa dimensions is associated with intense bone growth and osteoblastic activity at the skull sutures. From our measurements, the anterior cranial fossa undergoes an early phase of lengthwise growth (anisotropic growth), followed by

**Table 4. Coefficients of the segmented regression model of linear parameter changes as a function of fetal age: First segment:** $Y = a_0 + a_1 * Age$ **(***Age* **from 10 weeks to** $Age^{cut-off}$**), second segment:** $Y = b_0 + b_1 * Age$ **(***Age* **from** $Age^{cut-off}$ **to 26 weeks),** $a_0$ **–first segment constant,** $a_1$ **- slope of the first segment,** $b_0$ **–second segment constant,** $b_1$**—slope of the second segment),** $R$**—correlation coefficient.**

| Parameter $Y$ | $Age^{cut-off}$ (week) | $a_0$ | $a_1$ | $b_0$ | $b_1$ | $R$ |
|---|---|---|---|---|---|---|
| S-F (mm) | 18.0 | -9.7 mm | 1.93 mm/wk | 5.3 mm | 1.09 mm/wk | **_0.856_** |
| a (°) | 18.0 | 199.6° | -3.54°/wk | 112.0° | 1.31°/wk | **_0.407_** |
| h (mm) | 17.7 | -0.18 mm | 0.050 mm/wk | 0.43 mm | 0.016 mm/wk | **_0.646_** |
| hp (mm) | 17.5 | 1.00 mm | 0.047 mm/wk | 1.26 mm | 0.033 mm/wk | **_0.371_** |
| hm (mm) | 17.4 | 0.47 mm | 0.202 mm/wk | 0.80 mm | 0.182 mm/wk | **_0.383_** |
| hc (mm) | 17.0 | 0.23 mm | 0.087 mm/wk | 1.32 mm | 0.023 mm/wk | 0.169 |
| s (°) | 16.6 | 202.9° | -6.17°/wk | 91.4° | 0.53°/wk | **_0.630_** |
| FS-SA-SP (°) | 15.0 | 113.7° | 2.61°/wk | 158.1° | -0.29°/wk | 0.210 |

simultaneous length and width enlargement (isotropic growth). Subsequently, the geometry of the anterior cranial fossa was examined to compare its proportions with the other two cranial fossae. This was achieved using previously obtained distance measurements, which allowed for the triangulation of the skull base. Trigonometric relationships were also employed.

### 3.4. Summary of measurement results

From the conducted research, the process of enlarging the dimensions of the anterior cranial fossa does not occur uniformly. Initially, there is intensive growth in the anteroposterior dimension (allometric growth), while later, the growth rate in the longitudinal dimension of the anterior fossa aligns with the transverse dimension, and the anterior fossa then enlarges

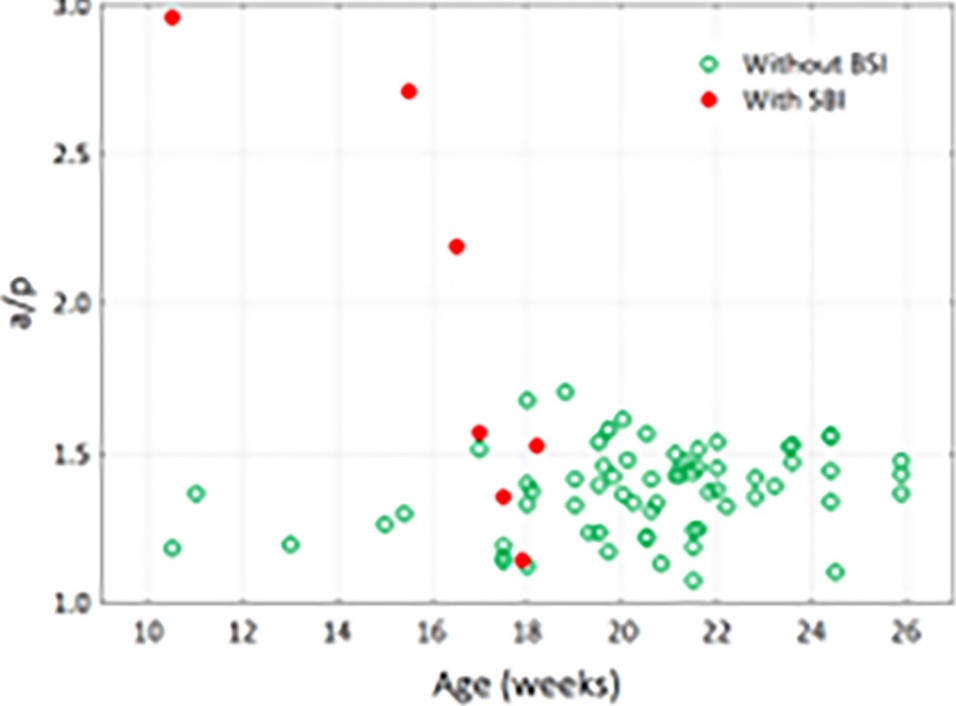

**Fig 8. Scatter plot of the a/p angle ratio depending on the age of 77 fetuses.**

**Table 5. Basic statistics (medians and quartiles) of distances and auxiliary angles on the left and right sides and results of significance tests (Wilcoxon test).**

| Symmetric parameters | Left side | Right side | *p*-value |
|---|---|---|---|
| Distance from the pituitary gland center to a zygomatic bone, *S-ZL* and *S-ZR* (mm) | 22.4 [19.2; 24.0] | 22.9 [19.9; 24.6] | ***0.011*** |
| Distance between the nasion point and a zygomatic bone, *F-ZL* and *F-ZR* (mm) | 28.6 [25.1; 31.1] | 29.9 [25.6; 32.2] | ***0.011*** |
| Distance between the S point and an ear cartilage, *S-PL* and *S-PR* (mm) | 27.4 [25.3; 30.9] | 28.0 [25.6; 30.0] | 0.132 |
| Distance between the ossification point Z and the point P, *ZL-PL* and *ZR-PR* (mm) | 26.4 [22.4; 28.1] | 25.2 [22.3; 27.7] | ***0.001*** |
| Angle of the anterior cranial fossa, *al* and *ar* (°) | 69 [65; 74] | 72 [66; 75] | 0.096 |
| Angle of the middle cranial fossa, *ml* and *mr* (°) | 60 [54; 66] | 58 [54; 64] | 0.204 |
| Angle of the posterior cranial fossa, *pl* and *pr* (°) | 51 [49; 53] | 51 [46; 52] | 0.688 |

while keeping proportions and angles of its parts (isometric growth). This is clear when comparing correlation diagrams of the S-F distance, which corresponds to the longitudinal dimension of the anterior fossa. In the midplane, there is a deepening of the cranial fossae with age, but particular attention is drawn to the dynamic increase in the depth of the posterior fossa, which consequently leads to a decrease in the angle of the cranial base, initially close to 180˚.

## 3.5. Skull structure and its function—drop-shaped reservoirs as a mechanical model of the skull

The primary function of the skull is to protect and support the brain located within its interior. Brain tissues are among the most hydrated tissues, immersed in cerebrospinal fluid. Therefore,

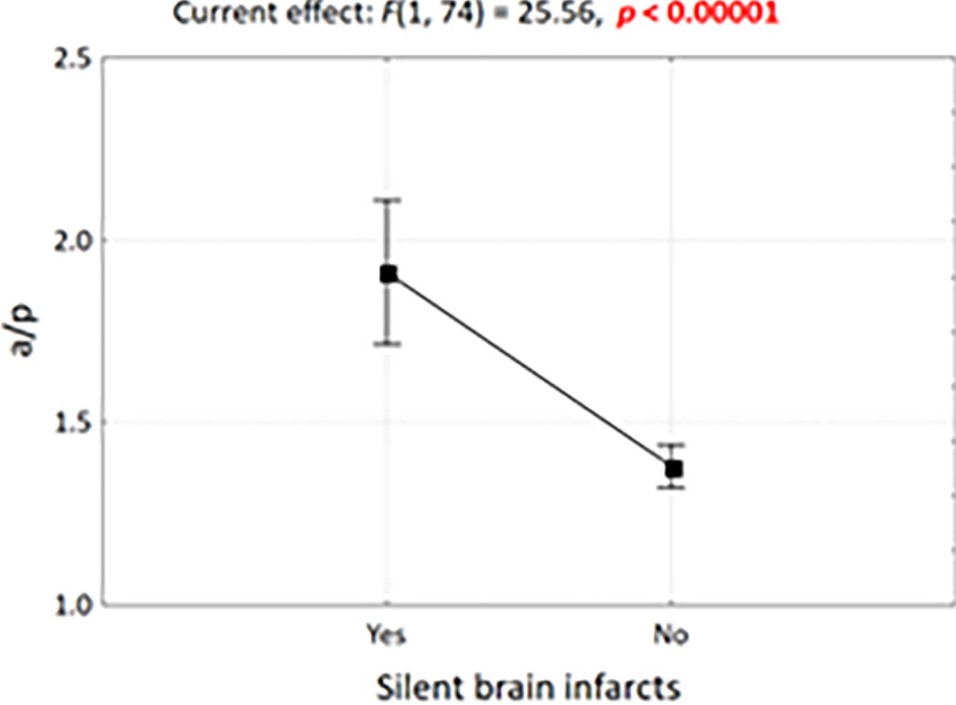

**Fig 9. Graphical interpretation of covariance analysis.** Expected marginal means of the a/p ratio for the SBI factor. Vertical bars indicate 0.95 confidence intervals.

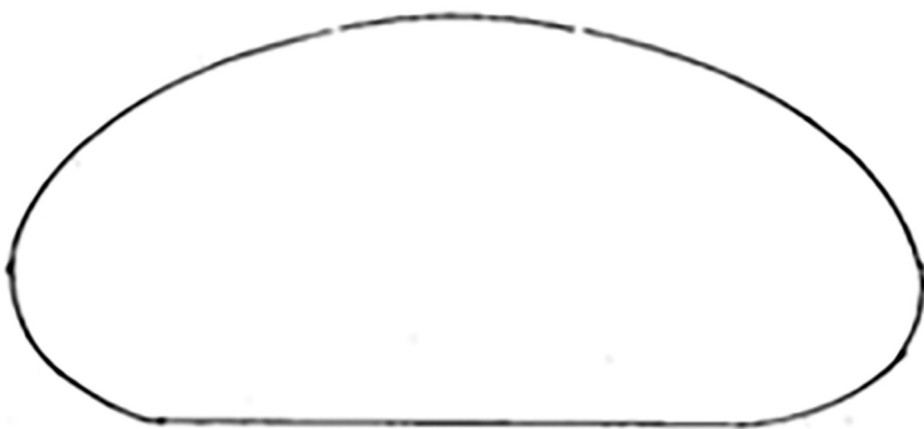

**Fig 10. Contour of a water droplet resting on a surface—cross-section base of drop-shaped reservoirs.**

the skull can be likened to a specific fluid reservoir for the brain. From a mechanical perspective, several types of reservoirs are distinguished, with drop-shaped reservoirs being the most proper model for the skull (Figs 10, 11). This group of reservoirs is characterized by a shape like that of a water droplet resting freely on the surface. Mathematically, the surface of such a reservoir is characterized, among other things, by equal stress vectors in all directions, particularly perpendicular directions at any given point on the surface. Already at the beginning of the 20th century, consideration was given to the possibility of using drop-shaped reservoirs for the storage of liquid fuels, shaped like deformed water droplets based on a hydrophobic surface. Such reservoirs, with minimal surface area, have the largest volume.

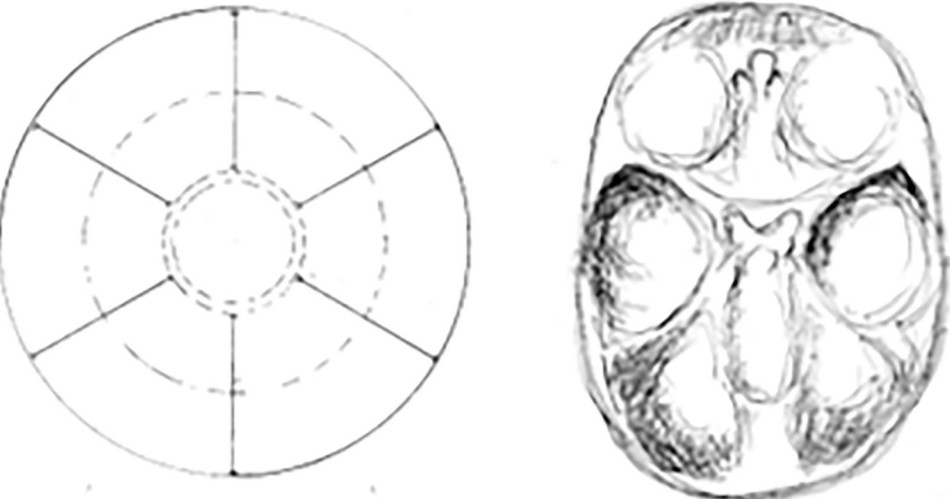

**Fig 11. Bottom view of reinforcing ribs strengthening the base of the reservoir; for comparison, a top view of the skull base (fetus with a vertex-sitz dimension of 198 mm).** Finally, the reinforcing rib system takes on a "mature" form. Further development will no longer significantly change the geometry and angles between the arches.

### 3.6. Characterization of changes during development and their clinical significance

Introduction to the characterization of the development of angles: anterior, middle, and posterior cranial fossae angles, the angle between the lesser wings of the sphenoid bone, and finally the angle of the cranial base significantly eases tracking changes in the geometry of the developing skull. During prenatal development, the skull behaves like an elevated drop-shaped reservoir from a mechanical perspective, associated with its function as a vessel containing cerebrospinal fluid and highly hydrated brain tissues. In the lower part of the skull reservoir, a rib system is formed to strengthen the structure of the shell, emanating concentrically from a single point into six ribs (reinforcing beams): one along each left and right pyramid, one along each left and right lesser wing of the sphenoid bone, one running from the body of the sphenoid bone anteriorly towards the ethmoid bone, and one running from the body of the sphenoid bone posteriorly along the slope towards the large occipital opening (Fig 11). The rib system reinforcing the base of the skull aims, after birth, to protect it and the brain contained within it from many head injuries, both trivial and frequent during a child's development, and more serious ones. Studies on the development of fetal skulls have shown that with age, ribs are formed in the base of the skull, stiffening a previously highly flexible shell.

These figures (Figs 12–20) describe the developmental stages and dimensions of the skull base in various fetal stages, illustrating changes and specific characteristics observed during prenatal growth.

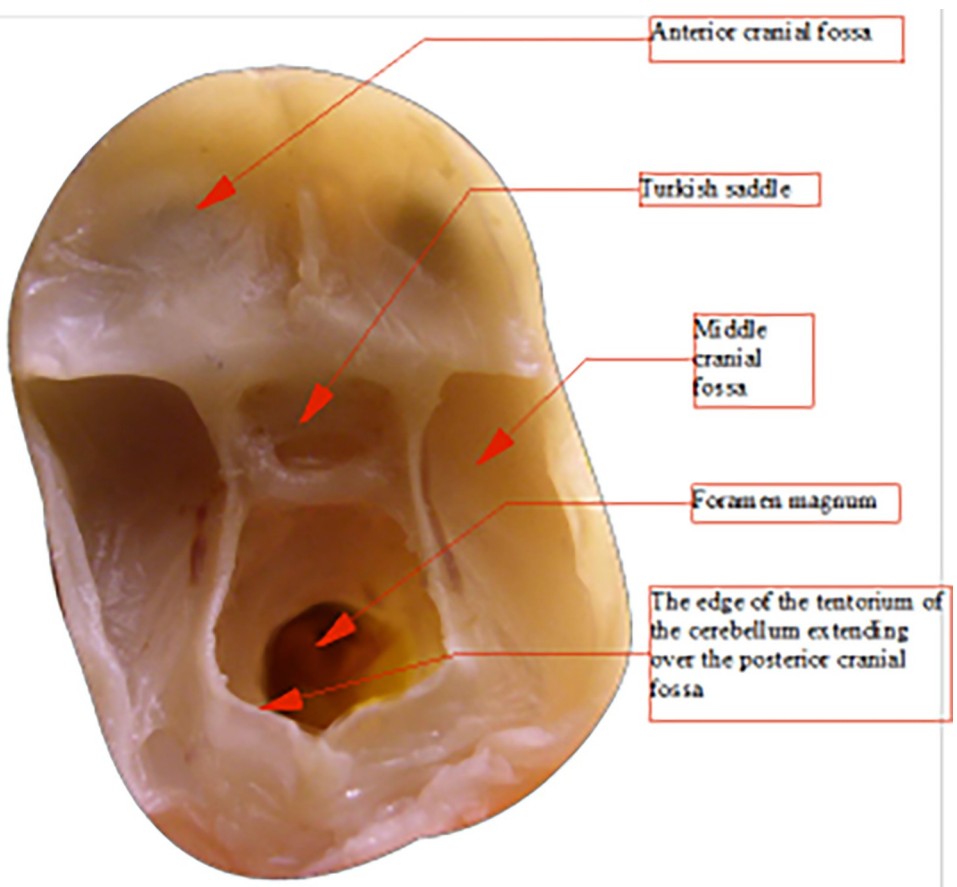

**Fig 12. Skull base, top view, fetus with occipitofrontal diameter 62 mm (age 10.5 weeks).**

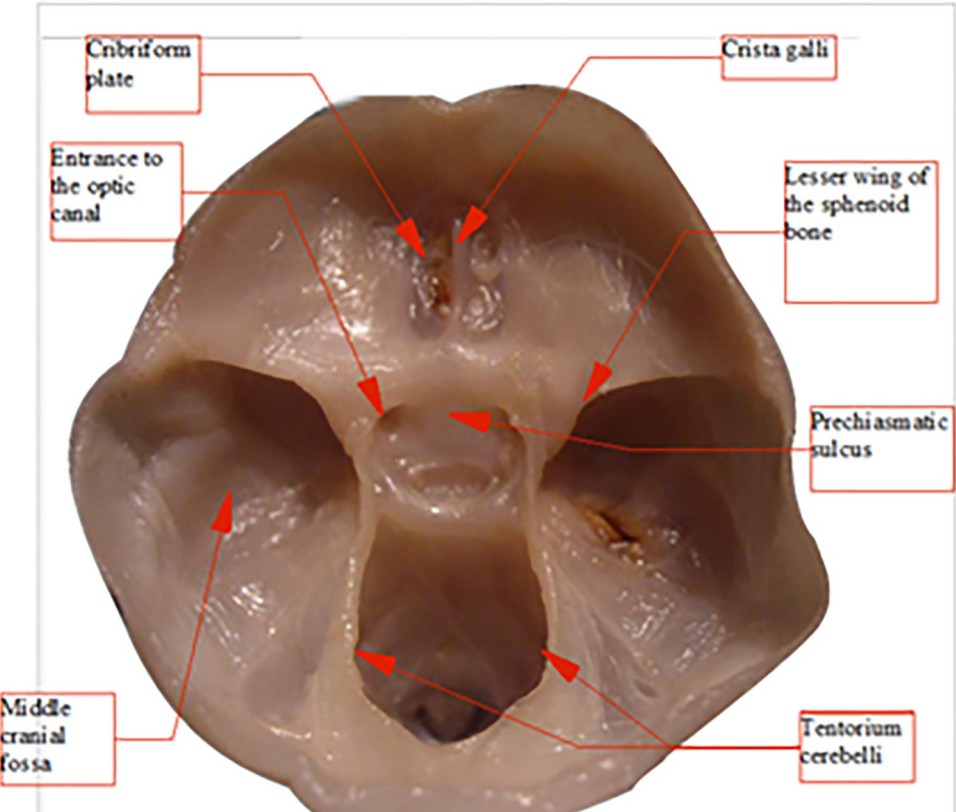

**Fig 13. Skull base, top view, fetus with occipitofrontal diameter 100 mm (age 13.5 weeks).**

## 4. Discussion

The quality of the fetal material from Wroclaw anatomical collections have been already described [22, 23].

Previous studies of the skull base development were conducted, among others, using radiographs of fetal bones. They led to the conclusion that the anterior skull fossa increases centrifugally towards the front, so that the angle of the anterior fossa at the vertex in the pituitary gland and the arms branching towards the points of ossification of the zygomatic bones remains constant during fetal development (isometric growth) [24]. Special attention should be paid to the latest, still relatively few neuroanatomical studies using fractal analysis [25, 26]. Analysis of the geometric relationships of the developing anterior fossa allows determination of the useful angle of the anterior skull base, with its vertex located in the pituitary gland, and the arms passing through the primary points of zygomatic bone ossification. On the arms of this angle, lying in a horizontal plane, the posterior edges of the lesser wings of the sphenoid bone are projected approximately (vertical projection). Whereas the lesser wings themselves form an angle in space, not lying in a horizontal plane.

From earlier studies, it follows that the size of this angle stays unchanged during fetal development [23]. However, both the studies of Korean authors and others [27–30] were performed using measurements on radiographs of fetal skulls, which influenced their accuracy. Modifications of these methods included studies using computer tomography [31, 32], or magnetic resonance imaging of formalin-fixed fetal corpses [33]. Radiological anatomy of the anterior fossa and skull base in the postnatal period is already quite well established [34, 35], unlike the

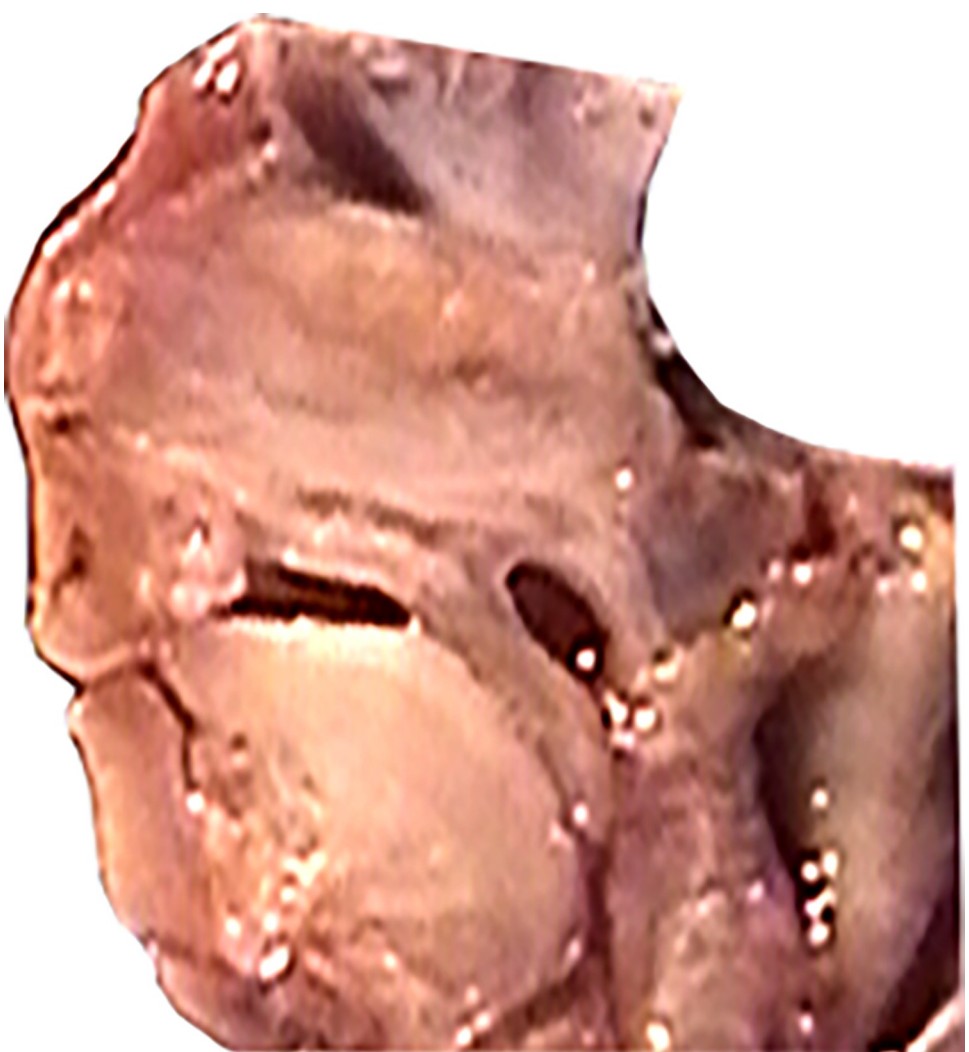

**Fig 14. Anterior cranial fossa, sagittal section, 15-week-old female fetus (5x magnification).**

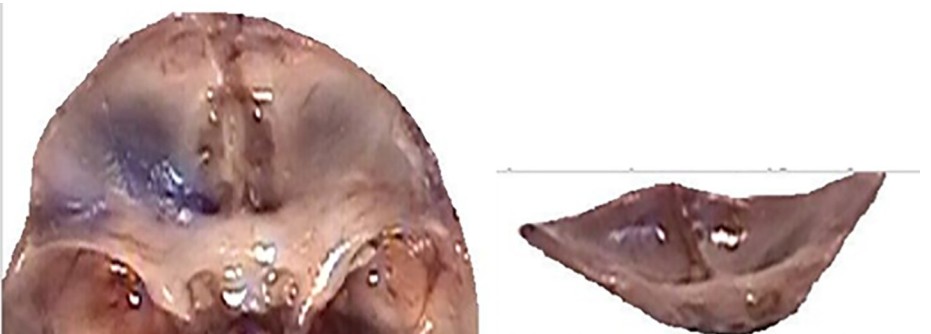

**Fig 15. Male fetus, age 17.5 weeks (3x magnification)- left; view of anterior cranial fossa from behind- right.**

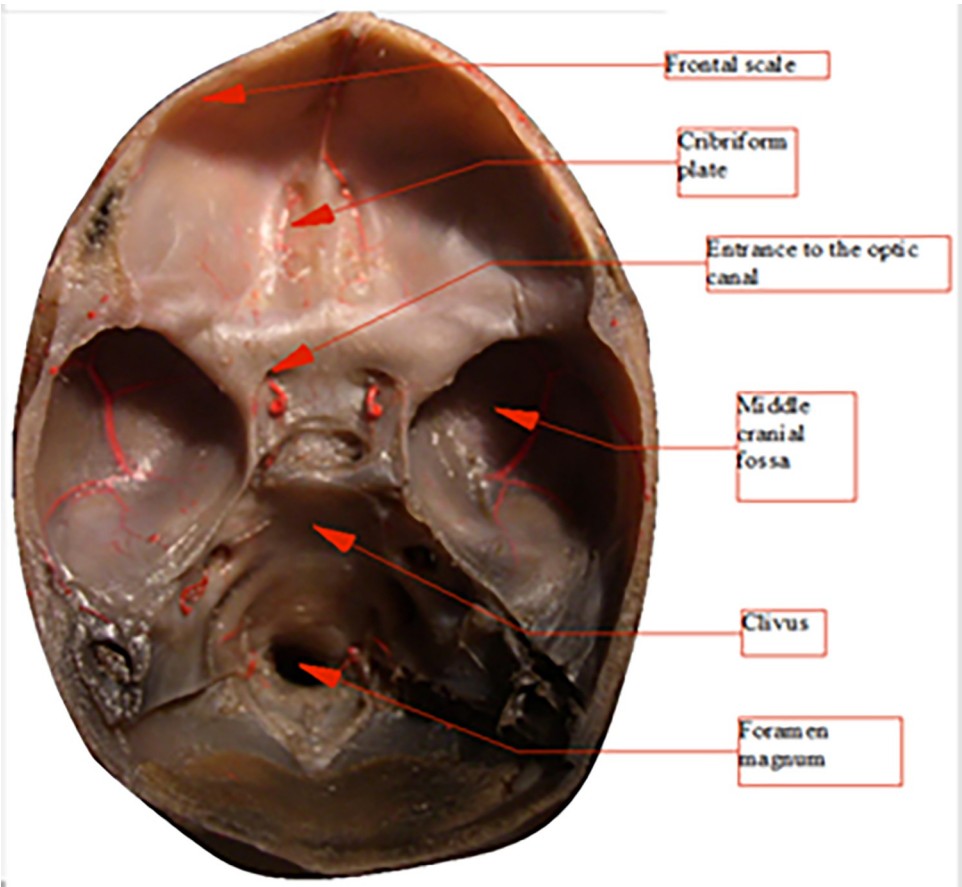

**Fig 16. Skull base, top view, fetus with occipitofrontal diameter 179 mm (age 19.5 weeks), injected blood vessels (red color).**

prenatal period. There are also many metrological studies of human skulls obtained from cadavers after birth [36, 37]. The study of sexual dimorphism of the human skull in the prenatal period is limited, unlike the well-studied dimorphism in adults [38–41]. Analysis of sexual dimorphism of the anterior skull base and its contents during prenatal development deserves attention. Comparative studies of male and female fetuses using ultrasound during pregnancy have shown slight advantages in the size of the lateral ventricles of the brain in males. However, the accuracy of the ultrasonographic method as well as studies using skull radiographs (above) did not allow definitive conclusions [42].

Some of the fetuses we examined had developmental abnormalities. In 20 cases there was visible pathology, areas of leukomalacia were found in 10 cases, intra- and periventricular hemorrhages in 5, and ventriculomegaly with colpocephaly in 2 fetuses. In one case of an 18-week-old fetus, lateral ventricular morphology typical of hydrocephalus was found, with ventricular triangles 8.5 mm wide. The frontal horns were most enlarged in both cases of hydrocephalus (100%) and were semicircular in nature, whereas after intraventricular and periventricular hemorrhages they were less enlarged and triangular, with the base of the triangle directed anteriorly and often with significant asymmetry [43].

In clinical practice, anatomical lines drawn on skull radiographs are used to diagnose impressions of the skull base (impressio basilaris): Chamberlain's line from the posterior edge of the hard palate to the posterior edge of the foramen magnum, McGregor's line from the

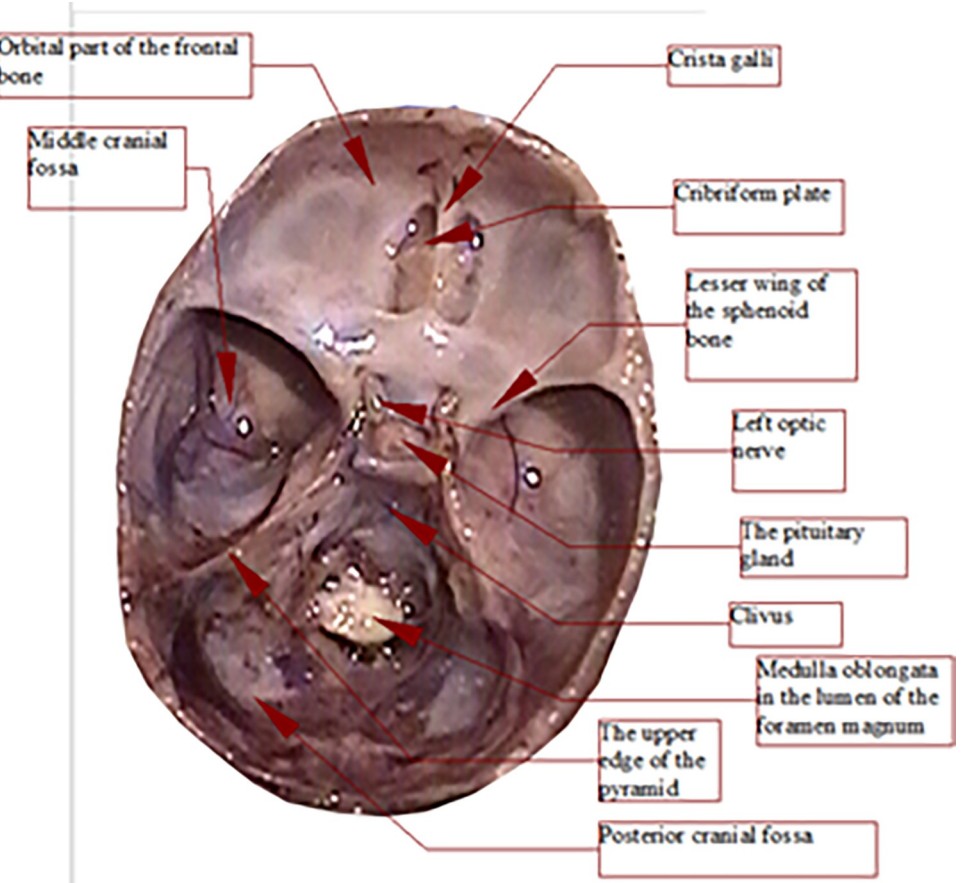

**Fig 17. Skull base, top view, fetus with occipitofrontal diameter 182 mm (age 19.5 weeks).**

posterior edge of the hard palate to the lowest point of the occipital bone, McRae's line—in the midline plane constituting the arrow dimension of the foramen magnum (from the basion point to the opisthion), Wackenheim's line from the saddle back to the lower edge of the slope (clivus), and Fischgold's. Recent neuroanatomical studies using fractal analysis provide many interesting, previously unknown pieces of information, for example, to describe brain vascularization and prenatal development of this vascularization[1].

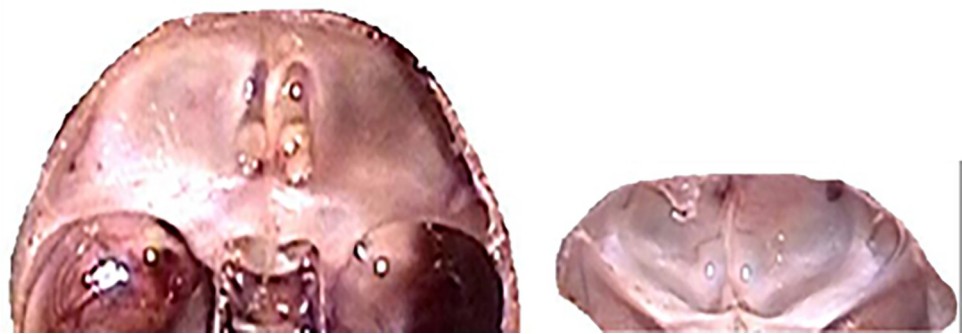

**Fig 18.** Anterior cranial fossa, top view (left); view of anterior cranial fossa from behind (right)- 19.5-week-old male fetus.

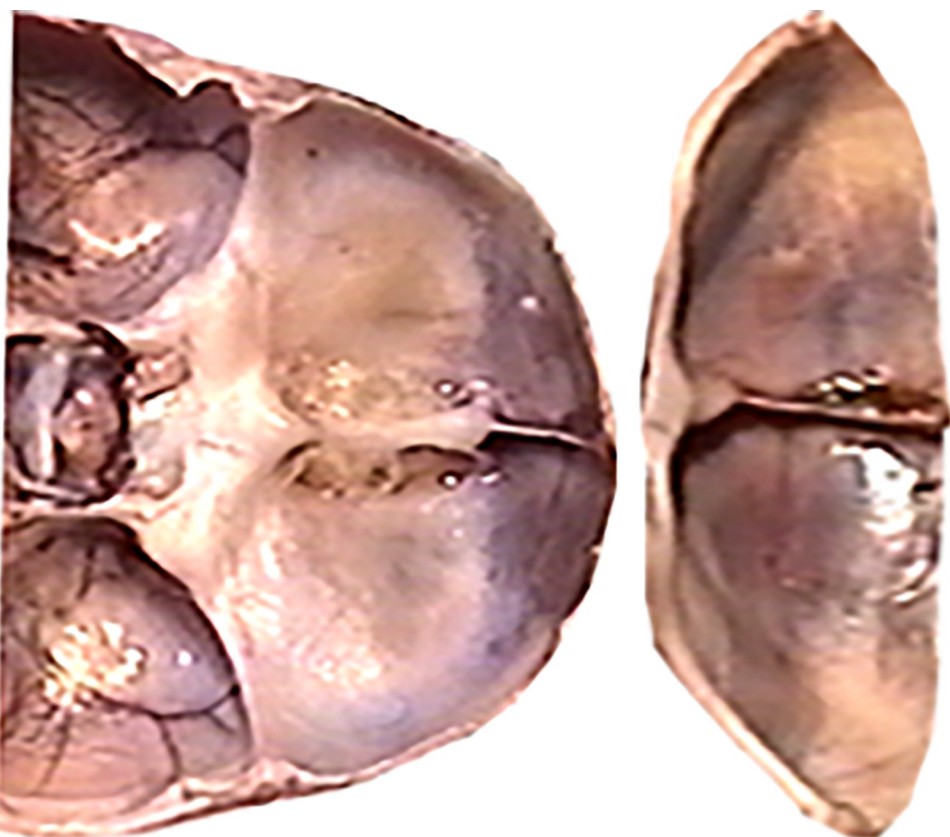

**Fig 19.** Anterior cranial fossa, 21-week-old female fetus (left); view from behind (right).

The examination of the skull base and its contents in patients after birth is currently the domain of imaging techniques: computer tomography and magnetic resonance imaging [44]. Interesting studies on plagiocephaly have been conducted using finite element analysis of computer tomographic scans of the skull [45]. Less often performed other studies include angiography [46]; in adults, angiography using contrast provides invaluable information about the course of brain vessels and their malformations [47]. Occasionally, single photon emission computed tomography–SPECT [48] is performed to assess cerebral blood flow. The facial skeleton structures are routinely evaluated in dentistry using panoramic radiographs [49]. However, these methods are not suitable for examining a fetus developing in the mother's womb. Only in recent years has there been hope for the application of ultrafast magnetic resonance imaging for prenatal imaging [50] Less invasive techniques such as ultrasonography are increasingly used in recent years, revealing many interesting findings [51, 52]. However, there is an urgent need to correlate neuroanatomical data obtained from ultrasonographic studies with direct measurements on fixed fetal specimens [53].

In recent years, there has been progress in information technology techniques, and the use of modern image processing methods allows for obtaining added information from analyzed images [54]. Our innovative method of computerized analysis of anatomical images allowed us to refine the data previously obtained using radiographs of fetal bones [55]. Metrological evaluation using computer analysis has already been successfully applied in other anatomical areas [56]. There have also been remarkably interesting studies of the venous system of the human brain in the prenatal period [57]. Using computed tomography and digital image analysis software ossification centers in the occipital bone were examined [58]. From the studies conducted

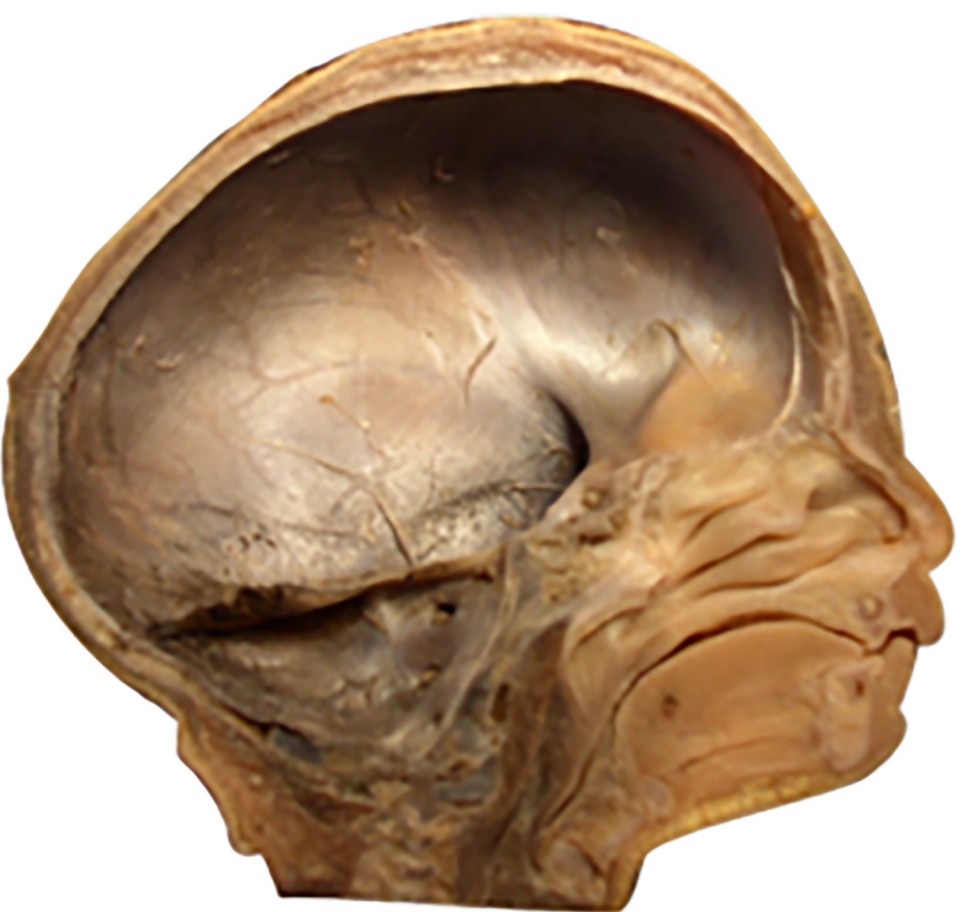

**Fig 20. Female fetus with v-pl 298mm, v-tub 208mm, sagittal section (age 25 weeks).**

on our entire fetal material, it follows that angle of the anterior cranial fossa, initially wide open, decreases during development. The discrepancy between our results and those of cited Korean authors [23] arises from a wider age range covered by our fetal material. The use of computer image processing techniques enabling distance measurement with an accuracy of one pixel is also important.

Our investigated dynamics of changes in skull base angles translate into uneven growth of the anterior cranial fossa in length and width (Figs 21, 22). Therefore, disorders associated with premature skull suture ossification will lead to deformity of the anterior cranial fossa shape, which occurs, for example, in craniosynostosis. Relating the changes in developing anatomical structure of the fetal skull base to the mechanical properties of capillary vessels leads to the conclusion that there is a close relationship between structure and function already in the prenatal period, which has not been the subject of research so far. The elements of reinforcement developing in the skull base during fetal development will shape into the supporting Felitzer's arches in adults:

- anterior (frontal-sphenoidal)

- two anterior-lateral (orbito-sphenoidal)

- two posterior-lateral (petro-squamous)

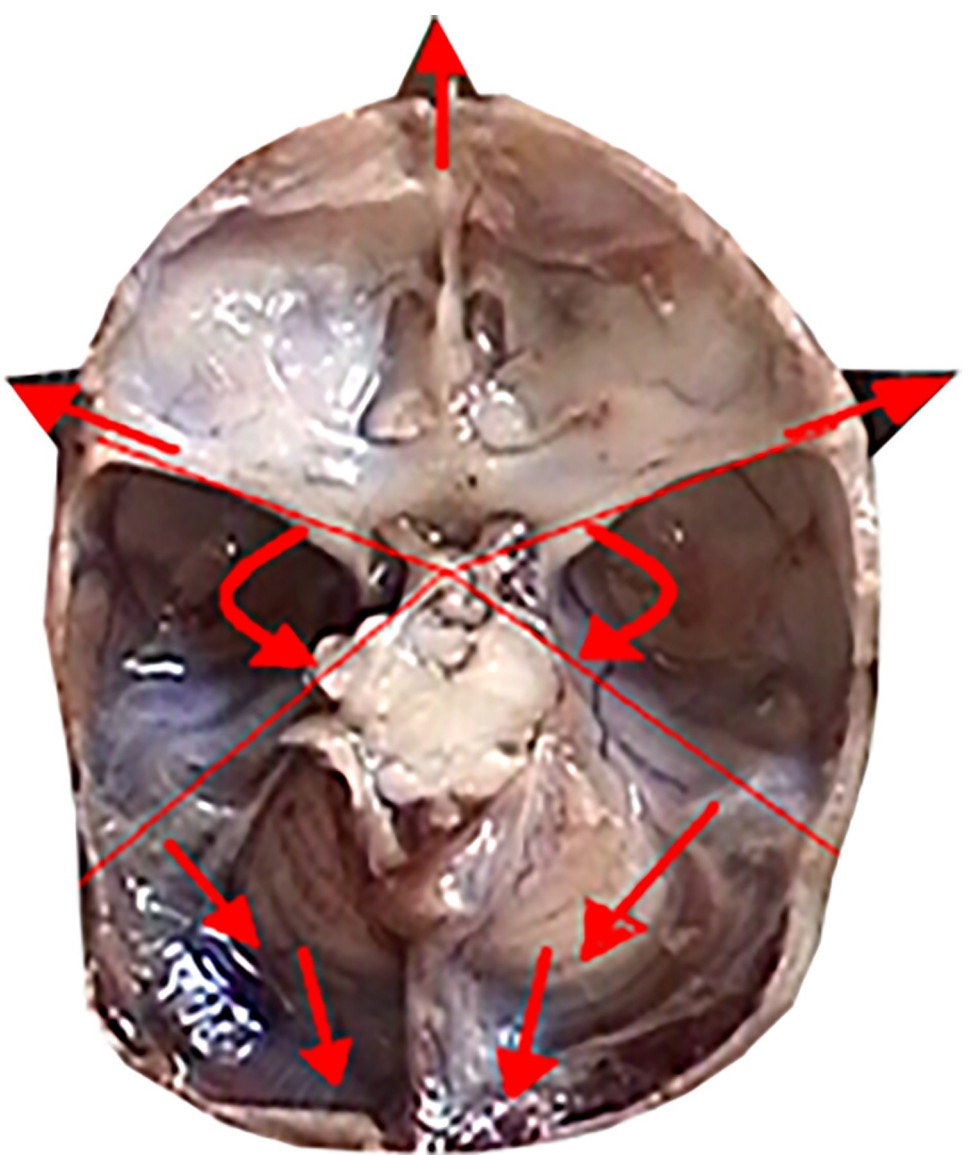

**Fig 21. Dynamics of changes in skull fossa angles.**

- posterior (occipital).

In our own material, the formation of reinforcing ribs in the skull base was seen early, from which the Felitzer's arches subsequently develop.

Drop-shaped reservoirs are according to our research the best mechanical model of the skull skull structure and its function.

Drop-shaped reservoirs can be classified, based on their installation method, into ground-level (based on a foundation seated on the ground) and elevated (supported, for example, on pillars and capable of serving as water towers). Both ground-level and elevated reservoirs can have two types of bottom parts: flat and spatial. The side of the reservoir is usually stiffened by appropriate ribs. Internal and external ribbing can be distinguished [59]. Drop-shaped reservoirs, particularly those elevated with internal ribbing on spatial bottoms, resemble the

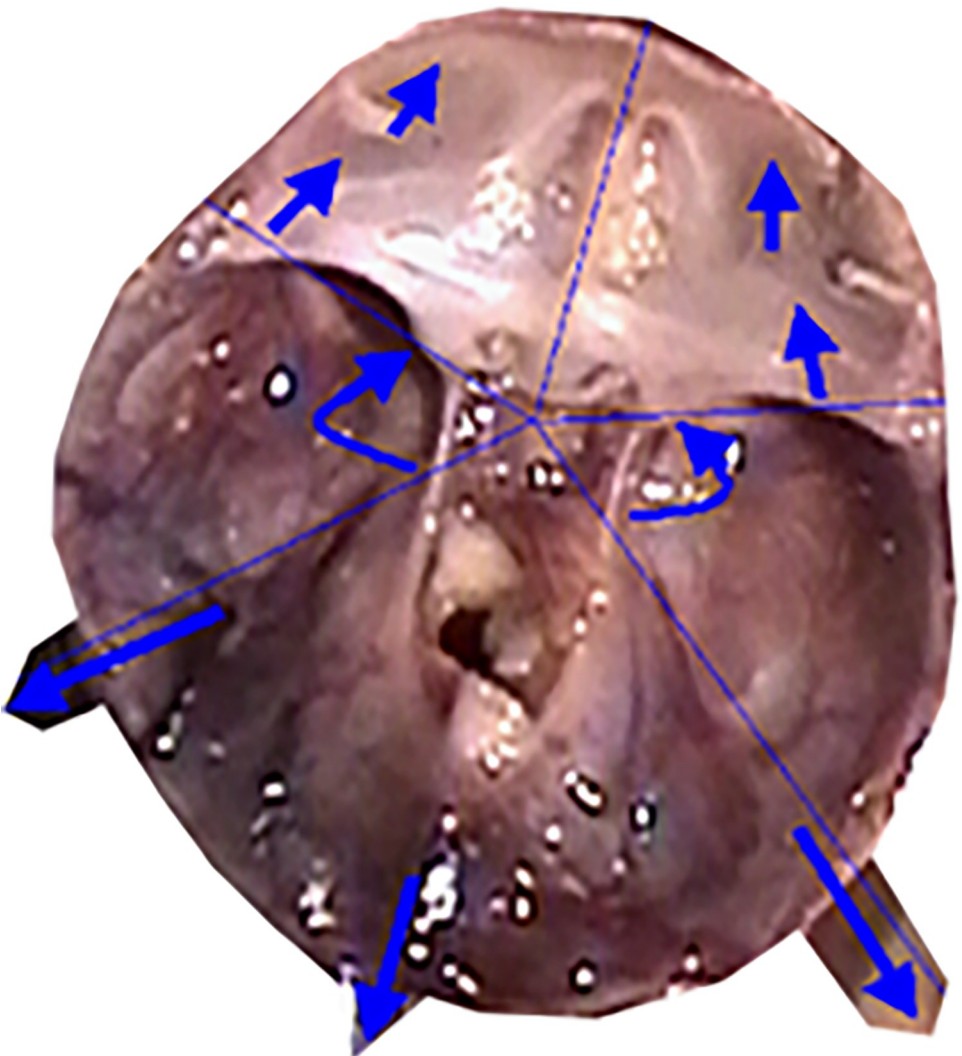

**Fig 22. Dynamics of changes in skull fossa angles earlier than in the previous drawing.**

construction of the human skull. Consideration of the conditions for the equilibrium of a water droplet resting on a hydrophobic surface leads to the conclusion that the values of surface and meridian stresses on the entire surface must be equal at each point. Such conditions are achieved only at constant pressure in the reservoir. In such a case, the drop-shaped reservoir is a shell of uniform strength, and thus of constant thickness. The condition of axial symmetry must also be met. Failure to meet any of these conditions necessitates the use of additional stiffening in the form of solid or lattice ribs. In cases where the design does not meet ideal assumptions regarding the load (liquid or liquid and gas pressure), proper shell shape, axial symmetry, thickness, or shell strength—surface stresses do not meet the accepted conditions, leading to a state of imbalance and requiring reinforcement through suitably selected ribbing. "Ribbing" here refers not to anatomical ribs, but rather structural elements used, among others, in architecture, resembling reinforcing arches supporting vaults. In liquid reservoirs, there are several areas weakened in the shell, such as those where control and technological hatches are located, pipes for filling and emptying the tank, and protective elements. These areas, in addition to reinforcement, are strengthened by appropriate thickening of the

shell [23]. Similar areas of weakening in the shell of the skull occur where blood vessels penetrate bone, nerve plexuses, as well as the location of organs of hearing and vision, or the connection of the cranial cavity with the spinal canal through the large occipital opening. Both in technical constructions and in the human skull, solutions aimed at reducing the weight of the object are implemented. An example includes using thinner shell walls in reinforced concrete reservoirs while simultaneously increasing the cross-section of reinforcement. An analogy to such an approach in the case of the skull is the lattice (openwork) structure of spongy bone. Elevated drop-shaped reservoirs can have external ribbing arranged in a pillar-shell system or an internal rib system connected to a suitably rigid ring. The internal rib system corresponds to the reinforcing system of the human skull [60]. Our anatomical analysis has important clinical implications. Surgical treatment of post-traumatic damage to the anterior cranial fossa is now a routine procedure in neurosurgery. However, precise anatomical and metrological evaluations of this region remain essential for refining current techniques and developing new approaches, especially in areas with challenging surgical access, such as the skull base, where tumors often occur. Detailed knowledge of neuroanatomical structures is crucial in reconstructive procedures, particularly in postoperative management of skull base tumors. For the potential improvement of existing neurosurgical procedures and the development of new ones, precise metrological assessment of skull anatomy is often crucial [61]. It is also essential for the development of new surgical methods in regions with difficult surgical access [62], which often occurs in cases of skull base tumors [63]. Special attention is needed for neuroanatomical relationships in reconstructive procedures within the skull base (e.g., postoperative treatment of tumors) [64]. For ophthalmological surgery in the orbital area, the presence of the cranio-orbital foramen (COF)—a hole in the lateral wall of the orbit leading to the anterior cranial fossa and found in approximately 48.37 percent of cases—may be important [65]. One of the most interesting congenital skull defects associated with asymmetry due to unilateral premature closure of sutures is plagiocephaly, which has been studied using postnatal computed tomography of the head [66].

Craniosynostosis, a condition characterized by premature ossification of cranial sutures, is a significant developmental defect that can severely limit skull growth. Depending on the sutures involved, it can lead to various deformities, such as brachycephaly (due to early closure of the coronal suture), scaphocephaly (sagittal suture), or oxycephaly (coronal and lambdoid sutures). Early diagnosis of craniosynostosis can enable neurosurgical intervention, allowing the creation of artificial sutures to correct the defect and promote normal skull growth. Recent studies, including animal models, have deepened our understanding of the biology behind suture ossification. These studies suggest that the dura mater plays a critical role in preventing premature closure, which opens the door to potential treatments aimed at modulating suture ossification. Although most research has focused on postnatal treatments, the possibility of addressing such defects prenatally is emerging. Early intervention during fetal development may one day prevent or mitigate the full spectrum of central nervous system disorders associated with these congenital anomalies, offering new hope for both diagnosis and treatment.

## 5. Conclusions

1. Changes in skull geometry: The growth of the anterior cranial fossa is not uniform. During the first trimester, there is allometric growth, with the longitudinal dimension increasing from 5 to 17 millimeters between the 8th and 14th week of fetal life. At the same time, the angle of the anterior cranial fossa decreases, and its depth increases towards the middle cranial fossa. In the second trimester, growth continues but becomes more uniform, with only slight changes in the angle of the anterior cranial fossa. There is a gradual decrease in the angle between the

lesser wings of the sphenoid bone as the depth of the anterior cranial fossa increases in the frontal plane.

2. Development of the anterior cranial fossa in relation to the other two cranial fossae: There is a significant increase in the depth of the posterior cranial fossa in the midplane, exceeding 21 millimeters by the end of the third trimester. The lowering of the posterior cranial fossa is accompanied by a decrease in the angle of the skull base, which initially is close to 180˚. The depth of the middle cranial fossa also increases from 2 millimeters in the 8th week to 5 millimeters in the 27th week of fetal life.

3. Symmetry: The development of the skull base within the examined age range occurs symmetrically relative to the body's midline.

4. Sexual dimorphism: Sexual dimorphism is evident in the area of the anterior cranial fossa even in the prenatal period. The angle of the anterior cranial fossa is greater in male fetuses than in female fetuses, while female fetuses exhibit a higher height of the crest of the sphenoid bone.

5. Clinical aspects and mechanical model of the skull: Many congenital defects are associated with abnormalities in the development of the skull base, as indicated by earlier animal experiments.

## Supporting information

**S1 Appendix.**
(DOCX)

## Author Contributions

**Conceptualization:** Wojciech Derkowski, Alicja Kędzia, Krzysztof Dudek.

**Data curation:** Wojciech Derkowski, Alicja Kędzia, Michał Glonek.

**Formal analysis:** Wojciech Derkowski, Alicja Kędzia, Krzysztof Dudek.

**Funding acquisition:** Wojciech Derkowski.

**Investigation:** Wojciech Derkowski, Alicja Kędzia, Michał Glonek.

**Methodology:** Wojciech Derkowski, Alicja Kędzia, Krzysztof Dudek, Michał Glonek.

**Project administration:** Wojciech Derkowski, Alicja Kędzia.

**Resources:** Wojciech Derkowski, Alicja Kędzia, Krzysztof Dudek, Michał Glonek.

**Software:** Wojciech Derkowski, Krzysztof Dudek.

**Supervision:** Wojciech Derkowski, Alicja Kędzia.

**Validation:** Wojciech Derkowski, Alicja Kędzia, Krzysztof Dudek.

**Visualization:** Wojciech Derkowski, Alicja Kędzia, Krzysztof Dudek.

**Writing – original draft:** Wojciech Derkowski, Alicja Kędzia, Krzysztof Dudek.

**Writing – review & editing:** Wojciech Derkowski, Alicja Kędzia, Krzysztof Dudek.

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
