## [Decision Letter · Decision Letter 0]

16 Sep 2024

PONE-D-24-32641Morphometric evaluation of the anterior cranial fossa during the prenatal stage in humans and its clinical implications.PLOS ONE

Dear Dr. Derkowski,

Thank you for submitting your manuscript to PLOS ONE. After careful consideration, we feel that it has merit but does not fully meet PLOS ONE’s publication criteria as it currently stands. Therefore, we invite you to submit a revised version of the manuscript that addresses the points raised during the review process.

We look forward to receiving your revised manuscript.

Kind regards,

Ryota Tamura

Academic Editor

PLOS ONE

Journal Requirements:

2. Please include a separate caption for each figure in your manuscript

Please include captions for your Supporting Information files at the end of your manuscript, and update any in-text citations to match accordingly. Please see our Supporting Information guidelines for more information: http://journals.plos.org/plosone/s/supporting-information.

Additional Editor Comments :

Major revision is needed.

Reviewers' comments:

Reviewer's Responses to Questions

**Comments to the Author**

1. Is the manuscript technically sound, and do the data support the conclusions?

Reviewer #1: Yes

Reviewer #2: Yes

2. Has the statistical analysis been performed appropriately and rigorously? 

Reviewer #1: Yes

Reviewer #2: Yes

3. Have the authors made all data underlying the findings in their manuscript fully available?

Reviewer #1: Yes

Reviewer #2: Yes

4. Is the manuscript presented in an intelligible fashion and written in standard English?

Reviewer #1: Yes

Reviewer #2: Yes

5. Review Comments to the Author

Reviewer #1: This is an interesting, multi-centre anatomical study using unique research material.

Due to the uniqueness of the material and the high relevance of the research problem, the work is worthy of evaluation and analysis by receivers.

Regarding the details of the work

ABSTRACT:

The abstract provides a detailed and thorough summary of the study, showcasing the use of innovative techniques and the clinical relevance of the findings. However, it would benefit from greater conciseness, clearer structure, and a more explicit presentation of the hypothesis and conclusions. Reducing some of the technical details and focusing on the most significant results and their implications would make the abstract more impactful and accessible.

Strengths:

1. Comprehensive Scope:

2. Innovative Techniques:

3. Clear Objectives and Methodology:

4. Clinical Relevance:

5. Detailed Findings:

Areas for Improvement:

1. Overly Detailed

2. Lack of Hypothesis or Research Question:

3. Technical Jargon:

4. Structure and Flow:

5. Conclusion and Implications:

The recommended modification of the abstract will make the paper more visible and increase the chances of the authors' results being of interest to the scientific world. The perfect abstract is the key to success.

Introduction

The introduction is thorough and provides a strong foundation for the research study, particularly in its detailed exploration of relevant congenital defects and developmental processes. However, it would benefit from some condensation and reorganization to enhance clarity and readability. My Recommendation: Make it shorter! Additionally, simplifying some of the technical language could make the introduction more accessible to a broader audience without losing its scientific rigor.

Methods:

“material characteristics”

I looked at the material with great interest. It came from a collection deposited in the anatomical unit . It would be useful to anonymise the data more: not to give the exact name of the unit and just describe it as a ‘local anatomical museum’.

I would also see more detailed information about this collection. I typed myself in pubmed with the keyword: ‘anatomical collections wroclaw’ and found 2-3 very good quality papers that refer to anatomical collections in wroclaw. I would recommend referring to these works. Describe this material and include papers confirming the quality of the material published in recent years, avoiding self-citation. The most interesting papers are those of Domanski J et al.

Other aspects of the methodology are very interesting and should arouse the interest of the readers.

In the results, I particularly like subsection 3. Relating observations related to skull growth to the drop shape reservoir is a very interesting way of analysing the data. Attempts to explain the mechanistic way in which the human skull develops are very interesting. I only have a formal question whether some of the sentences placed in this subsection should not be placed in the discussion chapter.

Discussion:

I would love to see a more detailed analysis for these sentences:

In contrast, the smaller wings themselves form an angle in space without lying in the horizontal plane. Previous studies suggest that the size of this angle remains unchanged during fetal development23. However, both the Korean authors' study and others27 28 29 30 were conducted using measurements on radiographs of fetal skulls, which affected their accuracy. Modifications of these methods have included studies using computed tomography31 32 or magnetic resonance imaging of formalin-fixed fetal cadavers33.

It would be interesting, but I do not know whether it would be possible to assess whether the observed fetuses individually have any abnormalities of cranial growth, which could theoretically lead sequentially to the clinical pathologies described in the discussion.

This part of the discussion concerning the clinical elements related to the anatomical analysis carried out needs to be somewhat revised and made more attractive.

Reviewer #2: Dear Authors

Thank you very much for submitting this highly interesting manuscript.

In their manuscript, the authors describe a morphometric study of the neurocranium of 77 human foetus specimens. A dissection of the fixed skulls and a surface scan using a video camera and image analysis software were performed. The following measurements were taken: Measurement features characterising the fetus, obtained from anthropometric measurements. Distances between specified measurement points in the anterior cranial fossa and other cranial fossae, obtained using computer image analysis programs. Derived features from measured distances, particularly values of specified angles characterising skull geometry.

The authors were able to show the different temporal growth stages of the anterior cranial fossa, the differences in sex-dependent development and the relationship to symmetry in body development. Furthermore, the group of authors assumes that their results can be used to better explain and possibly predict congenital malformations. They see clinical relevance with regard to the effects of surgical interventions in the area of the anterior skull base.

Overall, this is a clearly structured manuscript. The objectives are clearly formulated and the methodology is described comprehensively and comprehensibly. The results are adequately documented and supported and illustrated by meaningful images.

The discussion is comprehensive and takes into account the current literature.

It would be desirable to present an even clearer reference to clinical relevance, as this is a very theoretical anatomical question.

What concrete benefits does the clinician derive from these results?

6. PLOS authors have the option to publish the peer review history of their article (what does this mean?). If published, this will include your full peer review and any attached files.

Reviewer #1: No

Reviewer #2: **Yes: **Kai Johannes Lorenz

---

## [Author Response · Author response to Decision Letter 0]

19 Oct 2024

Opole, October 14, 2024

Dear Reviewers,

We would like to thank you very much for the work you put into reading our manuscript and for your very valuable comments that helped us improve it. Here are the answers to each point of the each review.

Reviewer #1:

"ABSTRACT: The abstract provides a detailed and thorough summary of the study, showcasing the use of innovative techniques and the clinical relevance of the findings. However, it would benefit from greater conciseness, clearer structure, and a more explicit presentation of the hypothesis and conclusions."

We have improved the structure of the abstract. Presentation of the hypothesis now is in its lines 3-6 and conclusions in lines 14-20. Because of point 5 beneath we decided maintain the lines 16-20.

"Reducing some of the technical details and focusing on the most significant results and their implications would make the abstract more impactful and accessible."

We have reduced significantly technical details in the abstract.

"Strengths:

1. Comprehensive Scope:

2. Innovative Techniques:

3. Clear Objectives and Methodology:

4. Clinical Relevance:

5. Detailed Findings"

We tried to maintain these points in the new version, so we decided maintain the lines 16-20.

"Areas for Improvement:

1. Overly Detailed"

We have reduced significantly technical details in the abstract.

"2. Lack of Hypothesis or Research Question"

Presentation of the hypothesis now is in its lines 3-6.

"3. Technical Jargon"

We have reduced significantly technical jargon.

"4. Structure and Flow"

We have improved the structure and flow of the abstract.

"5. Conclusion and Implications"

Conclusions is now in lines 14-20. Because of point 5 above we decided maintain the lines 16-20.

"Introduction

The introduction is thorough and provides a strong foundation for the research study, particularly in its detailed exploration of relevant congenital defects and developmental processes. However, it would benefit from some condensation and reorganization to enhance clarity and readability."

We have shortened and reorganized significant parts of the introduction to make it more clear and legible.

"My Recommendation: Make it shorter! Additionally, simplifying some of the technical language could make the introduction more accessible to a broader audience without losing its scientific rigor."

We have reduced significantly technical language and shortened whole introduction without, we believe, losing its scientific rigor.

"Methods:

“material characteristics”

I looked at the material with great interest. It came from a collection deposited in the anatomical unit. It would be useful to anonymise the data more: not to give the exact name of the unit and just describe it as a ‘local anatomical museum’."

We changed the name of the unit to ‘local anatomical museum’.

"I would also see more detailed information about this collection. I typed myself in pubmed with the keyword: ‘anatomical collections wroclaw’ and found 2-3 very good quality papers that refer to anatomical collections in wroclaw. I would recommend referring to these works. Describe this material and include papers confirming the quality of the material published in recent years, avoiding self-citation. The most interesting papers are those of Domanski J et al."

We have cited these articles (references 24 and 25): 

Domański J, Domagala Z, Simmons JE, Wanat M. Terra Incognita in anatomical museology - A literature review from the perspective of evidence-based care. Ann Anat. 2023 Jan;245:152013. doi: 10.1016/j.aanat.2022.152013. Epub 2022 Oct 17. PMID: 36257492.

 Domański J, Janczura A, Wanat M, Wiglusz K, Grajzer M, Simmons JE, Domagała Z, Szepietowski JC. Preservation fluids of heritage anatomical specimens - a challenge for modern science. Studies of the origin, composition and microbiological contamination of old museum collections. J Anat. 2023 Jul;243(1):148-166. doi: 10.1111/joa.13854. Epub 2023 Apr 6. PMID: 37024147; PMCID: PMC10273345.

"Other aspects of the methodology are very interesting and should arouse the interest of the readers."

We agree that research on the collection of fetuses from the Wrocław Department of Anatomy is very interesting, as our Department has been operating for over 100 years. However, it is not possible for us to discuss them even briefly - there are simply a huge number of these preparations. Our own photographic documentation from the autopsies of 77 fetuses alone includes over 30,000 photos (or rather frames from autopsy films, loaded into a computer and digitized). Therefore, we only allowed ourselves a mention at the beginning of the discussion chapter, with appropriate citations of articles on this topic.

"In the results, I particularly like subsection 3. Relating observations related to skull growth to the drop shape reservoir is a very interesting way of analysing the data. Attempts to explain the mechanistic way in which the human skull develops are very interesting. I only have a formal question whether some of the sentences placed in this subsection should not be placed in the discussion chapter."

We moved some of the sentences on human skull mechanics to the discussion chapter.

"Discussion:

I would love to see a more detailed analysis for these sentences:

In contrast, the smaller wings themselves form an angle in space without lying in the horizontal plane. Previous studies suggest that the size of this angle remains unchanged during fetal development23. However, both the Korean authors' study and others27 28 29 30 were conducted using measurements on radiographs of fetal skulls, which affected their accuracy. Modifications of these methods have included studies using computed tomography31 32 or magnetic resonance imaging of formalin-fixed fetal cadavers33."

We added short explanation in the end of 3rd paragraph of the discussion chapter.

"It would be interesting, but I do not know whether it would be possible to assess whether the observed fetuses individually have any abnormalities of cranial growth, which could theoretically lead sequentially to the clinical pathologies described in the discussion."

In the next paragraph, beneath fig.21 we described abnormalities leading to clinical pathologies from our study (we published it earlier - reference 45).

"This part of the discussion concerning the clinical elements related to the anatomical analysis carried out needs to be somewhat revised and made more attractive."

We changed some parts of the discussion for the new ones- especially last two paragraphs.

Reviewer #2:

"In their manuscript, the authors describe a morphometric study of the neurocranium of 77 human foetus specimens. A dissection of the fixed skulls and a surface scan using a video camera and image analysis software were performed. The following measurements were taken: Measurement features characterising the fetus, obtained from anthropometric measurements. Distances between specified measurement points in the anterior cranial fossa and other cranial fossae, obtained using computer image analysis programs. Derived features from measured distances, particularly values of specified angles characterising skull geometry.

The authors were able to show the different temporal growth stages of the anterior cranial fossa, the differences in sex-dependent development and the relationship to symmetry in body development. Furthermore, the group of authors assumes that their results can be used to better explain and possibly predict congenital malformations. They see clinical relevance with regard to the effects of surgical interventions in the area of the anterior skull base.

Overall, this is a clearly structured manuscript. The objectives are clearly formulated and the methodology is described comprehensively and comprehensibly. The results are adequately documented and supported and illustrated by meaningful images.

The discussion is comprehensive and takes into account the current literature.

It would be desirable to present an even clearer reference to clinical relevance, as this is a very theoretical anatomical question.

What concrete benefits does the clinician derive from these results?"

We have changed the last two paragraphs of the discussion to show the specific benefits that patients have already gained and may gain in the future from thoroughly understanding and describing in detail the development of the human anterior cranial fossa during the fetal period.

We hope that the corrections we have made will be sufficient.

Kind regards

Wojciech Derkowski

---

## [Decision Letter · Decision Letter 1]

4 Nov 2024

Morphometric evaluation of the anterior cranial fossa during the prenatal stage in humans and its clinical implications.

PONE-D-24-32641R1

Dear Dr. Derkowski,

We’re pleased to inform you that your manuscript has been judged scientifically suitable for publication and will be formally accepted for publication once it meets all outstanding technical requirements.

Kind regards,

Ryota Tamura

Academic Editor

PLOS ONE

Additional Editor Comments (optional):

Reviewers' comments:

Reviewer's Responses to Questions

**Comments to the Author**

1. If the authors have adequately addressed your comments raised in a previous round of review and you feel that this manuscript is now acceptable for publication, you may indicate that here to bypass the “Comments to the Author” section, enter your conflict of interest statement in the “Confidential to Editor” section, and submit your "Accept" recommendation.

Reviewer #1: All comments have been addressed

Reviewer #2: All comments have been addressed

2. Is the manuscript technically sound, and do the data support the conclusions?

Reviewer #1: Yes

Reviewer #2: Yes

3. Has the statistical analysis been performed appropriately and rigorously? 

Reviewer #1: Yes

Reviewer #2: Yes

4. Have the authors made all data underlying the findings in their manuscript fully available?

Reviewer #1: Yes

Reviewer #2: Yes

5. Is the manuscript presented in an intelligible fashion and written in standard English?

Reviewer #1: Yes

Reviewer #2: Yes

6. Review Comments to the Author

Reviewer #1: I am fully satisfied with your very good quality answers! In my opinion that paper meets with PLOS ONE criteria for good publication!

Reviewer #2: Dear Authors,

alls remarks had been taken to account, thanks alot for your work.

It is a very technical paper focused on a very specific topic, but it gives , for some specialist, some new informations.

7. PLOS authors have the option to publish the peer review history of their article (what does this mean?). If published, this will include your full peer review and any attached files.

Reviewer #1: No

Reviewer #2: **Yes: **Kai Johannes Lorenz

---

## [Editor Report · Acceptance letter]

8 Nov 2024

PONE-D-24-32641R1 

PLOS ONE

Dear Dr. Derkowski, 

I'm pleased to inform you that your manuscript has been deemed suitable for publication in PLOS ONE. Congratulations! Your manuscript is now being handed over to our production team.

Kind regards, 

on behalf of

Dr. Ryota Tamura 

Academic Editor

PLOS ONE